# Somatic driver mutation prevalence in 1844 prostate cancers identifies *ZNRF3* loss as a predictor of metastatic relapse

Michael Fraser[1,2,3 ✉], Julie Livingstone [4,5,6,7], Jeffrey L. Wrana [8,9], Antonio Finelli[1,2,10], Housheng Hansen He[10,11], Theodorus van der Kwast [12,13], Alexandre R. Zlotta[1,2,8,9], Robert G. Bristow [11,14,15] & Paul C. Boutros [4,5,6,7,11,16 ✉]

Driver gene mutations that are more prevalent in metastatic, castration-resistant prostate cancer (mCRPC) than localized disease represent candidate prognostic biomarkers. We analyze 1,844 localized (1,289) or mCRPC (555) tumors and quantify the prevalence of 113 somatic driver single nucleotide variants (SNVs), copy number aberrations (CNAs), and structural variants (SVs) in each state. One-third are significantly more prevalent in mCRPC than expected while a quarter are less prevalent. Mutations in *AR* and its enhancer are more prevalent in mCRPC, as are those in *TP53*, *MYC*, *ZNRF3* and *PRKDC*. *ZNRF3* loss is associated with decreased ZNRF3 mRNA abundance, WNT, cell cycle & PRC1/2 activity, and genomic instability. *ZNRF3* loss, RNA downregulation and hypermethylation are prognostic of metastasis and overall survival, independent of clinical and pathologic indices. These data demonstrate a strategy for identifying biomarkers of localized cancer aggression, with *ZNRF3* loss as a predictor of metastasis in prostate cancer.

[1] Department of Surgery, Division of Urology, University of Toronto, Toronto, ON, Canada. [2] Department of Surgical Oncology, Genitourinary Site Group, Princess Margaret Cancer Centre, Toronto, ON, Canada. [3] Ontario Institute for Cancer Research, Toronto, ON, Canada. [4] Department of Human Genetics, University of California, Los Angeles, CA, USA. [5] Department of Urology, University of California, Los Angeles, CA, USA. [6] Institute for Precision Health, University of California, Los Angeles, CA, USA. [7] Jonsson Comprehensive Cancer Centre, University of California, Los Angeles, CA, USA. [8] Lunenfeld Research Institute, Mount Sinai Hospital, Toronto, ON, Canada. [9] Mount Sinai Hospital, Sinai Health System, Toronto, Canada. [10] Princess Margaret Cancer Centre, Toronto, ON, Canada. [11] Department of Medical Biophysics, University of Toronto, Toronto, ON, Canada. [12] Department of Laboratory Medicine and Pathobiology, University of Toronto, Toronto, ON, Canada. [13] Laboratory Medicine Program, University Health Network, Toronto, ON, Canada. [14] Manchester Cancer Research Centre, Manchester, UK. [15] University of Manchester, Manchester, UK. [16] Department of Pharmacology & Toxicology, University of Toronto, Toronto, ON, Canada. ✉email: michael.fraser@cpcgene.com; PBoutros@mednet.ucla.edu

While the vast majority of prostate cancers are organ-confined at diagnosis[1], a significant proportion of these tumors relapse following surgery or radiation therapy[2,3]. This necessitates salvage therapy to prevent or limit the development of distant metastases[4]. For example, up to 20% of men with intermediate-grade prostate cancer will experience biochemical relapse within three years of definitive local therapy[5,6], which portends an aggressive clinical course. There is therefore a clear need to improve on current risk-stratification guidelines, which rely on three clinical prognostic factors (i.e. Gleason/ISUP grade, pre-treatment serum concentration of prostate-specific antigen (PSA), and clinical T category).

Many somatic mutations have been proposed to predict relapse of localized prostate cancer[7–14]. However, the majority of these studies focus on weak surrogates of disease-specific mortality (e.g. biochemical relapse; BCR). Moreover, while genome-wide analysis offers an unbiased approach to biomarker discovery, discovering, characterizing, and validating these biomarkers is limited by the high false discovery rate that results from simultaneous testing of multiple mutations for association with clinical outcome. Mutations associated with the lethal disease are evolutionarily selected for localized tumors[15]. This suggests that mutations that are highly prevalent in lethal metastatic, castration-resistant prostate cancer (mCRPC) but rare in localized disease may be prognostic biomarkers that reflect the elevated risk of occult metastatic disease. While there are some data comparing the prevalence of driver mutation prevalence in localized disease vs. mCRPC[16,17], a comprehensive analysis of the clinical impact of this differential is lacking.

To address these gaps, we quantify the prevalence of 113 mutation types in 72 established prostate cancer driver genes or recurrently mutated genomic regions[7,18] in 1844 patients with either localized prostate cancer or mCRPC. We identify the differential prevalence of seventy-three established driver mutations, including mutations in the androgen receptor and its enhancer region. Amongst these differentially prevalent driver mutations, twenty-four are present in at least 5% of localized cancers and four are significantly associated with metastatic relapse of localized disease, including genomic gains in *MYC* and *CCND1*. In addition, genomic loss of the WNT pathway inhibitor *ZNRF3* is associated with WNT pathway activity and predicts biochemical and metastatic relapse and overall survival in localized prostate cancer. These results demonstrate a method of identifying candidate prognostic genomic biomarkers by comparing primary and metastatic disease and establish *ZNRF3* loss as a predictor of aggressive localized prostate cancer.

## Results

**Molecular drivers of localized and metastatic prostate cancer.**
We collected somatic SNV, CNA, and SV calls from eleven DNA sequencing studies of prostate cancer comprising 1844 patients (1289 localized, 555 metastatic; Fig. S1 and S2 and Supplementary Data 1)[7,11–15,18–22]. We curated a list of 113 mutation types—31 CNAs, 43 coding SNVs, 6 non-coding SNVs and 33 SVs—from those identified as either putative drivers or recurrently mutated in two large whole-genome sequencing studies of prostate cancer[7,18]. These encompassed 72 individual genes or genomic loci (Supplementary Data 2).

The most common mutation in localized prostate cancers was loss of *NKX3-1*, in 644/1279 cases (50.4%; Fig. 1 and Supplementary Data 3). Other mutations present in at least 20% of localized cancers were ERG SVs (78/201 cases; 38.8%), PTEN SVs (45/201 cases; 22.4%), MYC gains (267/1279 cases; 20.9%), and CDH1 losses (256/1279 cases; 20.0%). The most common mutation in mCRPC was gain of the androgen receptor

(*AR*), which occurred in 395/555 cases (71.2%; Fig. 1 and Supplementary Data 3). Consistent with the higher rate of mutation reported in mCRPC relative to localized disease[16], 25 genes were mutated in at least 20% of mCRPC cases, compared with only 4 in localized prostate cancer (Fig. 1 and Supplementary Data 3).

The proportion of localized prostate cancer and mCRPC cases harboring each driver mutation is shown in Fig. 2A. To assess which mutations are more prevalent in each disease state, we first evaluated the difference in these proportions ('observed Δ proportion'). Because of the differences in mutational burden between localized disease and mCRPC[18,22], we also derived an expected Δ proportion based on 100,000 samples of the binomial distribution, per driver gene mutation, per sample, weighted by mutational burden and gene size, in each tumor sample (Fig. 2B; see "Methods"). Finally, we computed the difference between observed Δ proportion and expected Δ proportion, yielding an adjusted Δ proportion which indicates whether a driver mutation is prevalent in mCRPC more than expected (adjusted Δ proportion > 0) or less than expected (adjusted Δ proportion < 0), based on background global mutation burden.

Overall, 37.2% of driver gene mutations (42/113) were significantly more prevalent than expected in mCRPC ($q < 0.05$, adjusted Fisher's Exact test; Fig. 2C and Supplementary Data 3)[21,23]. By comparison, 27.4% (31/113) were significantly less prevalent. Driver mutations more prevalent in mCRPC may either provide a selective advantage to metastasis when localized or may result from adaptation to metastatic niches and response to therapy. The largest adjusted Δ proportion was for genomic gain of the *AR* locus (gene body or enhancer), which was present in 395/555 (71.2%, 95% CI: 67.4–74.9) mCRPCs but only 2/1279 (0.16%; 95% CI: 0–0.37) localized cancers (adjusted Δ proportion = 0.609, 95% CI: 0.587–0.632, $q = 2.35 \times 10^{-5}$, adjusted Fisher's Exact test; Fig. 2C). SNVs in *AR* were less common overall and were not observed in localized disease (76/555 vs. 0/1204; adjusted Δ proportion = 0.128, 95% CI: 0.113–0.144, $q = 2.35 \times 10^{-5}$, adjusted Fisher's Exact test). CNAs in *BRCA2* were frequently observed and significantly more prevalent in mCRPC than expected (219/555 vs. 72/1279; adjusted Δ proportion = 0.232, 95% CI: 0.212–0.251, $q = 2.35 \times 10^{-5}$, adjusted Fisher's Exact test), consistent with reports showing that germline and somatic aberrations in *BRCA2* are more prevalent in mCRPC than localized disease[24,25]. CNAs and SNVs in *TP53* were both significantly more prevalent in mCRPC (SNV: 205/555 vs. 49/1204; adjusted Δ proportion = 0.305, 95% CI: 0.284–0.332, $q = 2.35 \times 10^{-5}$; CNA: 336/555 vs. 156/1279; adjusted Δ proportion = 0.426, 95% CI: 0.403–0.448, $q = 2.35 \times 10^{-5}$, adjusted Fisher's Exact test). Similarly, SVs affecting *TP53* were more prevalent in metastatic disease (adjusted Δ proportion$_{SV}$ = 0.074, 95% CI: 0.044–0.104, $q = 0.060$, adjusted Fisher's Exact test). While this did not reach our pre-set statistical significance threshold, it is important to note the lower power to detect differences in SV prevalence, relative to other mutational classes. Consistent with previous reports[16,26], *SPOP* SNVs were less prevalent in mCRPC[16] (32/555 vs. 103/1204; adjusted Δ proportion = −0.051, 95% CI: −0.061–0.041; $q = 2.35 \times 10^{-5}$, adjusted Fisher's Exact test).

**Candidate genomic biomarkers of aggressive localized prostate cancer.** Relapse of localized prostate cancer following curative-intent therapy is due, at least in part, to the presence of occult metastatic disease at initial presentation. We hypothesized that mutations prevalent in mCRPC than localized disease might be prognostic for relapse of localized prostate cancer. From the list of 73 mutations with significantly different prevalence in mCRPC

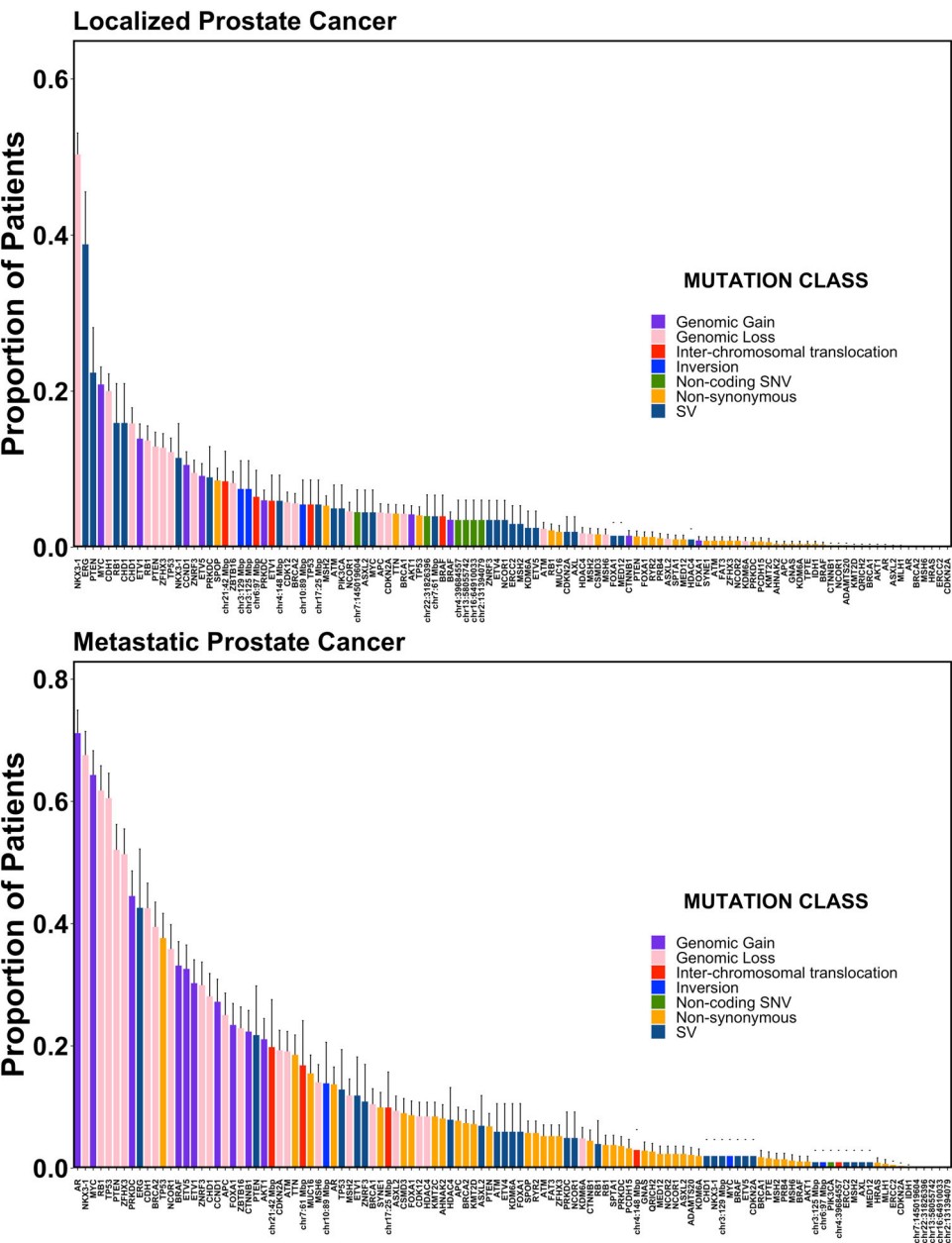

**Fig. 1 Frequency distribution of genes affected by driver mutations in prostate cancer.** The full cohort ($n = 1844$) was split into localized ($n = 1289$) and metastatic ($n = 555$) disease. Estimated proportion of patients in each cohort harboring each of 113 specific driver mutations. *X*-axis indicates the gene or genomic locus affected. Bar color indicates the type of driver mutation affecting that gene. Several genes are affected by multiple mutation types (e.g. *TP53*, *PTEN*, and others). Bars show the proportion of patients harboring the indicated mutation. Error bars represent 95% confidence intervals. Centre of the error bars represent the observed proportion. Source data are provided as a Source Data file.

vs. localized disease (42 with increased prevalence, 31 with lower prevalence), we identified 24 that occurred in at least 5% of all localized cancers evaluated (Supplementary Data 4). These 24 driver mutations comprised 16 CNAs, 6 SVs, and 2 SNVs. We then used univariable Cox proportional hazards modeling to assess whether any of these 24 mutations are associated with metastatic relapse of localized disease. Using patient outcome data from the CPCG study ($n = 376$), 4 of the 24 driver mutations were both more prevalent in mCRPC and associated with significantly higher risk of metastatic relapse ($q < 0.05$; Table S1): *MYC*, *CCND1*, and *PRKDC* gain & *ZNRF3* loss. Three other mutations—losses of *CDK12*, *ETV5*, and *TP53*—were associated with metastatic relapse but did not reach our pre-determined level of statistical significance ($q < 0.05$). *PTEN* and *RB1* loss were

not prognostic of metastatic relapse (although *PTEN* loss was associated with biochemical relapse in the CPCG cohort; $p = 0.035$, log-rank test). In a multivariable Cox proportional hazards model including all seven driver CNAs with $q \leq 0.25$, *ZNRF3* loss, *MYC* gain, *CCND1* gain, and *CDK12* loss remained significantly associated with risk of metastatic relapse (Fig. S3).

To confirm these associations with adverse outcome, we employed two independent validation cohorts: TCGA PRAD[19] and MSKCC[27] localized prostate cancer cohorts. In TCGA, *MYC* and *CCND1* gains as well as *ZNRF3* losses were prognostic for progression-free survival (Table S2); on multivariable analysis, only *ZNRF3* remained prognostic (Table S2). Similarly, these CNAs were also prognostic for poor outcomes in the MSKCC cohort (Tables S3, S4). Taken together, these data support the

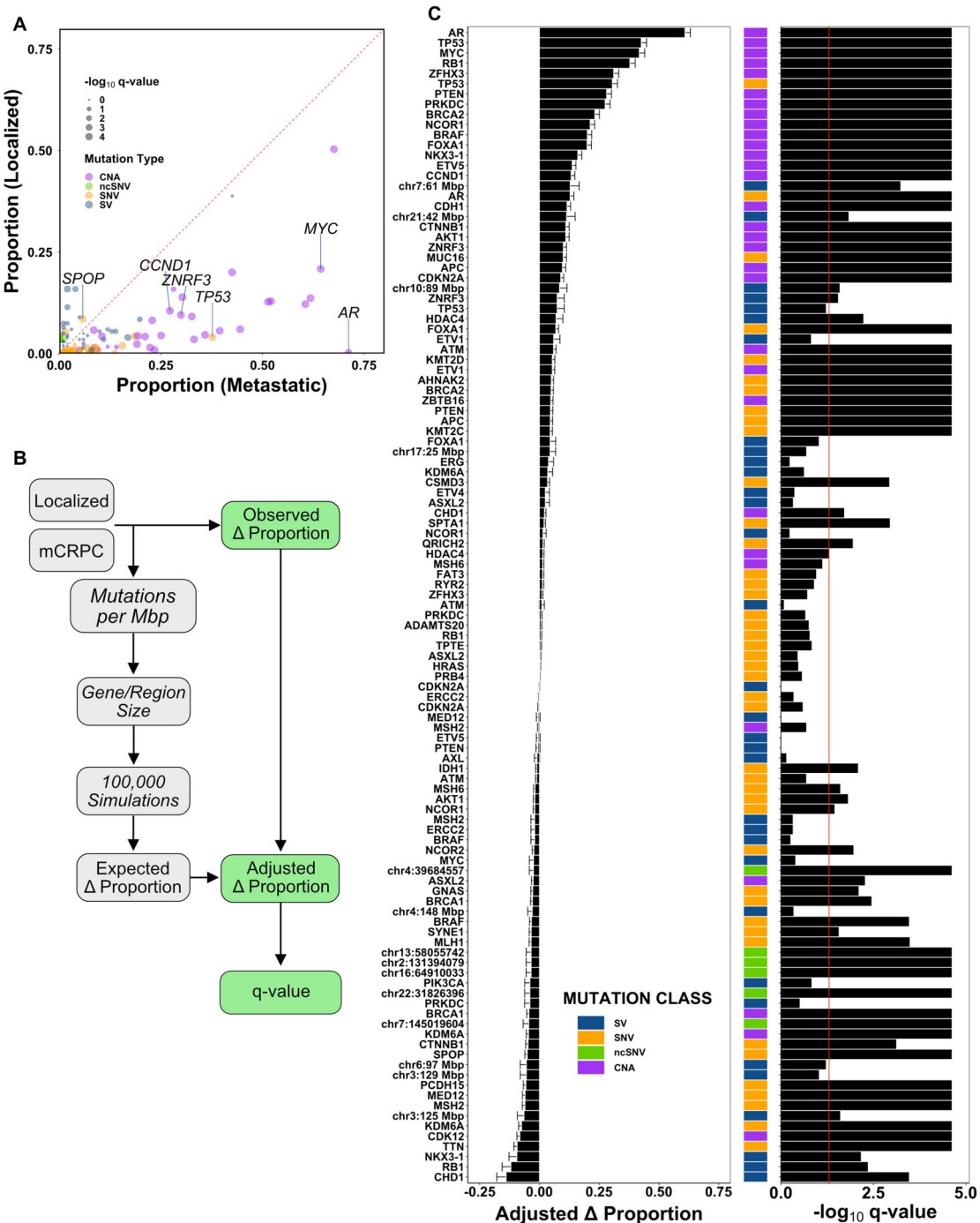

**Fig. 2 Prevalence of driver gene mutations in localized and metastatic prostate cancer. A** The proportion of tumors harboring each driver mutation in localized prostate cancer or mCRPC ('Observed Δ Proportion'), as described. Dot size indicates −log₁₀ *q*-value; dot color indicates driver mutation type. Specific genes of interest are labeled. **B** Comparison of driver gene mutation prevalence in localized disease and mCRPC. Differences in proportion of localized and metastatic cases harboring each driver mutation ('Observed Δ Proportion') were subtracted from the difference in proportions resulting from 100,000 simulations, per gene, per sample, where the probability of observing a mutation in a given sample was weighted by the global mutational burden (i.e. SNVs per Mb or PGA) in that sample ('Expected Δ Proportion') to generate the Adjusted Δ Proportion. A two-sided *p*-value was calculated as the proportion of simulated proportions that were as extreme or more extreme than the observed Δ proportion. *q*-values were then derived using the False Discovery Rate method. **C** Positive adjusted Δ proportion indicates higher than expected prevalence in mCRPC, while negative adjusted Δ proportion indicates lower than expected prevalence. Mutations are ordered from top to bottom by adjusted Δ proportion, as shown by the height of each bar. Error bars represent 95% confidence intervals; centre of the error bars represents the adjusted Δ proportion. Statistical significance was tested using adjusted Fisher's Exact tests with correction for multiple testing using the False Discovery Rate method. Mutations with *q*-value < 0.05 (red line) were considered statistically significant. Source data are provided as a Source Data file.

utility of this approach to prognostic biomarker discovery. *MYC*, *CCND1*, and *PRKDC* have previously been linked to adverse clinical outcomes in prostate cancer[28–31], and thus serve as positive controls. *ZNRF3* loss has been reported in five mCRPC patients[18,22] (two monoallelic and three biallelic) and several studies have observed losses in the region containing *ZNRF3* (chr22q12)[7,19,27,32,33]. To the best of our knowledge, however, ZNRF3 loss has not previously been explicitly reported as a recurrent CNA in localized prostate cancer or associated with adverse clinical outcomes in localized disease.

**ZNRF3 genomic loss is associated with gene expression and clinical features**. We next focused on the functional and clinical implication of *ZNRF3* loss. *ZNRF3* loss was identified in 166/555 metastases (29.9%; 95% CI: 26.1–33.7) and 122/1279 localized cancers (9.54%; 95% CI: 7.93–11.1; adjusted Δ proportion = 0.10, 95% CI: 0.086–0.113, $q = 2.35 \times 10^{-5}$, adjusted Fisher's Exact test). While *ZNRF3* loss was present in ~10% of localized cancers overall, this CNA was enriched in aggressive disease. In CPC-GENE, 25% of patients (9/36) who relapsed metastatically harbored *ZNRF3* loss while only 5.2% of patients (18/341) who did not harbor it (OR = 5.98, 95% CI: 2.45–14.6; $p = 3.29 \times 10^{-4}$, Fisher's Exact test; Fig. S4A). Similarly, in the TCGA cohort, *ZNRF3* loss was identified in 11.9% (58/489) of patients; of the 91/489 patients with disease progression, 20/91 (22.0%) harbored *ZNRF3* loss while only 9.55% of patients who did not progress harbored this CNA (OR = 2.67, 95% CI: 1.47–4.85; $p = 1.64 \times 10^{-3}$, Fisher's Exact test; Fig. S4B). *ZNRF3* loss was also associated with higher grade tumors in both CPCG and TCGA (Fig. S4C). Thus, *ZNRF3* loss appears to identify a subtype of localized prostate cancer associated with aggressive clinical outcomes.

*ZNRF3* loss was prognostic for BCR and metastatic relapse on univariable Cox proportional hazards analysis in CPCG ($HR_{BCR} = 2.18$, 95% CI: 1.31–3.64, $p = 2.87 \times 10^{-3}$, Wald test; $HR_{METS} = 4.57$, 95% CI: 2.12–9.84, $p = 1.03 \times 10^{-4}$; Wald test; Fig. 3A, B). These effects remained significant after controlling for standard clinical prognostic variables (i.e. pre-treatment PSA, diagnostic ISUP grade group, and clinical T category) in multivariable Cox proportional hazards modeling ($HR_{BCR} = 1.77$, 95% CI: 1.05–3.00, $p = 0.034$, Wald test; $N_{BCR} = 126/376$; $HR_{METS} = 3.01$, 95% CI: 1.35–6.74, $p = 7.32 \times 10^{-3}$, Wald test; $N_{METS} = 36/376$; Fig. 3C, D; univariable Cox models for all clinical prognostic factors are shown in Table S5).

To validate the transcriptomic impact of *ZNRF3* genomic loss, we assessed ZNRF3 RNA abundance in 208 CPCG tumors with matched RNA abundance and CNA data. ZNRF3 RNA abundance spans a limited range across tumor specimens (mean $\log_2$ RNA abundance = 6.63, range: 6.09–7.47; Fig. S5). Tumors harboring a genomic loss in *ZNRF3* had significantly lower ZNRF3 RNA abundance than *ZNRF3* neutral tumors (mean $\log_2$ RNA abundance difference = 0.163, 95% CI: 0.132–0.195, $p = 0.021$, Mann–Whitney U test; Fig. S5). This association validated in TCGA (mean $\log_2$ RNA abundance difference = 0.450, 95% CI: 0.303–0.596, $p = 4.06 \times 10^{-6}$, Mann–Whitney U test; Fig. S5) and Taylor/MSKCC cohorts (mean $\log_2$ RNA abundance difference = 0.075, 95% CI: 0.070–0.079, $p = 1.13 \times 10^{-2}$, Mann–Whitney U test; Fig. S5). Only four patients in the Gerhauser cohort with *ZNRF3* loss had available RNA abundance data, but there was a trend toward decreased ZNRF3 RNA abundance relative to ZNRF3 neutral patients (mean $\log_2$ RNA abundance difference = 1.37, 95% CI: −1.42 to 4.16, $p = 0.11$, Mann–Whitney U test; Fig. S5).

ZNRF3 RNA abundance was inversely associated with ISUP grade in four of five independent cohorts of localized prostate cancer cohorts (Fig. S6). Likewise, in CPCG, ZNRF3 RNA abundance was inversely associated with risk of BCR on multivariable analysis controlling for clinical ISUP grade, pre-treatment PSA, and clinical T-category ($N_{BCR} = 46/205$; $HR_{BCR} = 0.22$, 95% CI: 0.06–0.86, $p = 0.030$, Wald test; Fig. S7A). ZNRF3 RNA abundance was not significantly associated with metastatic relapse on multivariable analysis, potentially due to a lack of statistical power resulting from the low event rate in this intermediate risk cohort ($N_{METS} = 16/205$ patients; $HR_{METS} = 0.20$, $p = 0.18$, Wald test). ZNRF3 RNA abundance remained inversely associated with BCR risk after controlling for *ZNRF3* loss in multivariable Cox proportional hazards analysis ($HR_{BCR} = 0.21$, 95% CI: 0.05–0.78, $p = 0.02$, Wald test; $N_{BCR} = 46/205$; Fig. S7B).

We validated this association between ZNRF3 RNA and risk of adverse clinical outcome in three independent cohorts: TCGA, EOPC, and a high risk/high-volume intermediate risk cohort (LTRI[34]), in which low ZNRF3 RNA abundance was associated with a significantly higher risk of PFS, BCR, and metastatic relapse (Fig. S8). In the LTRI cohort, 18/18 patients with low ZNRF3 RNA abundance experienced metastatic relapse within 7 years following surgery, compared with only 2/29 patients with high ZNRF3 RNA abundance ($p = 4.16 \times 10^{-11}$, Fisher's Exact test; Fig. S8D). ZNRF3 RNA was not related to the risk of disease progression in the Taylor/MSKCC cohort. Taken together, these data demonstrate that genomic loss or low RNA abundance of *ZNRF3* preferentially occur in aggressive localized prostate cancer, independent of standard clinical prognostic factors.

**ZNRF3 loss predicts poor outcome localized prostate cancer**. We recently developed a six-feature clinico-genomic signature that predicts biochemical relapse in men with localized prostate cancer[7]. To assess whether *ZNRF3* genomic loss adds independent prognostic value to these features, we stratified the CPCG cohort based on both signature features and *ZNRF3* loss. Overall, 298/379 (78.6%) CPCG cases had informative data for the six features in the signature (*MYC* gain, *ATM* SNVs, *TCERGL1* hypomethylation, *ACTL6B* hypermethylation, chr7:61 Mbp inter-chromosomal translocations, and clinical T category). On multivariable analysis, both the Fraser signature and *ZNRF3* remained prognostic of BCR and metastasis, after controlling for ISUP grade group, PSA, and clinical T category ($HR_{BCR} = 1.77$, 95% CI: 1.06–2.98, $p = 0.030$, Wald test; $N_{BCR} = 126/375$; $HR_{METS} = 2.86$, 95% CI: 1.29–6.34, $p = 0.019$, Wald test; $N_{METS} = 36/375$; Table S6).

In the CPCG cohort, *ZNRF3* genomic loss was associated with a significantly higher percentage of the genome altered by a CNA (Mean Adjusted PGA: 6.96%, 95% CI: 6.27–7.66 vs. 11.1%, 95% CI: 7.66–14.5, $p = 4.57 \times 10^{-3}$, Mann–Whitney U test; Fig. S9A), and this validated in the TCGA (Mean Adjusted PGA: 13.2%, 95% CI: 11.5–15.0 vs. 30.0%, 95% CI: 24.0–36.1, $p = 2.11 \times 10^{-10}$, Mann–Whitney U test; Fig. S9B) and Taylor/MSKCC cohorts (Mean Adjusted PGA: 27.8%, 95% CI: 14.2–41.4 vs. 9.45%, 95% CI: 7.51–11.4, $p = 6.40 \times 10^{-4}$, Mann–Whitney U test; Fig. S9C). Despite this, *ZNRF3* genomic loss remained prognostic of metastatic relapse in the CPCG cohort after correcting for adjusted PGA, PSA, ISUP grade, clinical T category, and intraductal carcinoma of the prostate/cribriform architecture (IDC-P/CA)—an established negative prognostic factor[35,36] in multivariable Cox proportional hazards models ($HR_{METS} = 3.53$, 95% CI: 1.47–8.47, $p = 4.72 \times 10^{-3}$, Wald test; $N_{METS} = 30/326$; Table 1). In contrast, *ZNRF3* loss was not prognostic of BCR after correcting for these clinical and pathologic prognostic features ($HR_{BCR} = 1.49$, 95% CI: 0.83–2.67, $p = 0.180$, Wald test; $N_{BCR} = 106/326$). These results support the hypothesis that *ZNRF3* loss is

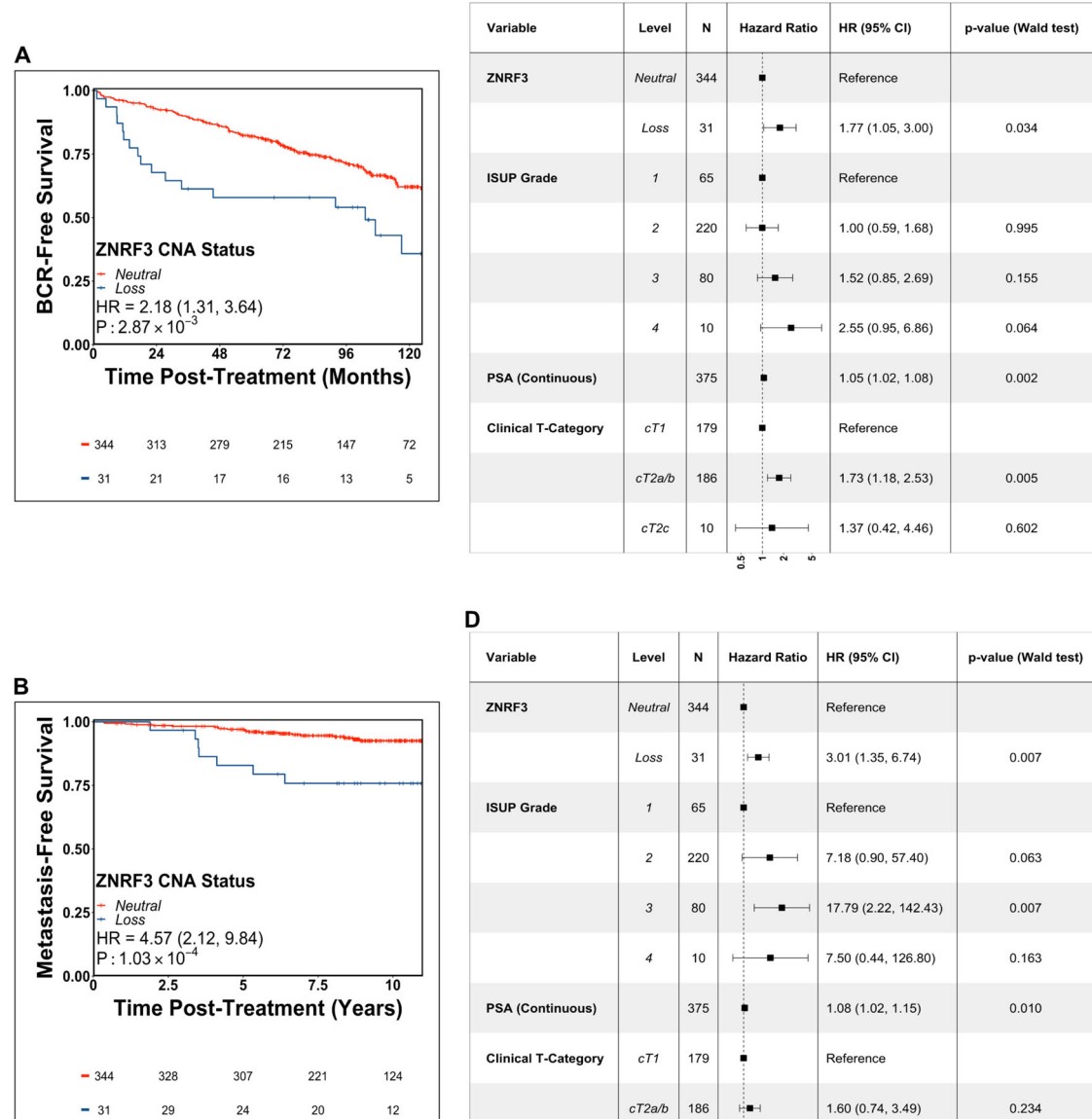

**Fig. 3 ZNRF3 genomic loss is an independent prognostic factor for aggressive localized prostate cancer.** Biochemical relapse-free rate (**A**) and metastatic relapse-free rate (**B**; mRFR) in CPCG patients with tumor specimens with (blue) or without (red) genomic loss of *ZNRF3*. Two-sided *p*-values generated from a Wald test. **C, D** Forest plots of multivariable Cox proportional hazards analyses of *ZNRF3* CNA status with clinical prognostic factors for biochemical relapse (**C**) and metastatic relapse (**D**). Error bars indicate 95% confidence interval of the reported hazard ratio. Source data are provided as a Source Data file.

an independent predictor of metastatic relapse in localized prostate cancer.

*ZNRF3* loss occurred in the context of a relatively large region of genomic loss on chromosome 22q12.1 (median deletion: 6.0 Mbp, range: 1.01–35.1 Mbp) and the smallest number of genes co-deleted with *ZNRF3* was 10 (median co-deleted genes = 90), covering 1.31 Mbp. All 417 genes on chromosome 22 were co-deleted with *ZNRF3* in at least one patient. Of these, 29 (5.3%) were significantly associated with metastatic relapse in CPCG ($q < 0.01$, Wald test; Table S7). Of these 29 genes co-deleted with *ZNRF3* and associated with metastasis, 9/29 also showed a significant reduction in RNA abundance in cases with genomic loss in this region ($q < 0.05$, Mann–Whitney U test), and thus represent likely candidate drivers of the aggressive phenotype

associated with this deletion. To further refine this gene list and help to identify the driver of aggression in this region, we next determined the association between RNA abundance of these 9 genes and adverse clinical outcomes in four independent cohorts (CPCG, TCGA, EOPC, and LTRI) using univariable Cox proportional hazards models (Table S8). The only gene with RNA abundance significantly associated with adverse outcome across all four cohorts ($q < 0.05$) was *ZNRF3*. These data are thus consistent with the hypothesis that *ZNRF3* loss drives clinical aggression of localized prostate cancer.

**Molecular Hallmarks of ZNRF3 genomic loss**. To further understand the functional correlates of *ZNRF3* in prostate

**Table 1 Multivariable Cox proportional hazards model of ZNRF3 loss on metastatic relapse, corrected for clinical prognostic factors, adjusted PGA, and IDC-P/CA.**

| Variable | Level | N | Hazard Ratio | 95% Confidence Interval | *p*-Value |
|---|---|---|---|---|---|
| ZNRF3 Loss | | | | | |
| | Neutral | 298 | – | – | – |
| | Loss | 28 | 3.53 | 1.47–8.47 | 4.72E−03 |
| PSA | Continuous | 326 | 1.05 | 0.96–1.14 | 0.31 |
| Gleason Score | | | | | |
| | 3+3 | 54 | – | – | – |
| | 3+4 | 190 | 4.29 | 0.53–34.6 | 0.17 |
| | 4+3 | 73 | 9.62 | 1.15–80.3 | 0.036 |
| | 4+4 and above | 9 | 5.33 | 0.31–93.0 | 0.25 |
| Clinical T Category | | | | | |
| | cT1 | 157 | – | – | – |
| | cT2a/b | 160 | 1.63 | 0.68–3.94 | 0.27 |
| | cT2c | 9 | 2.47 | 0.29–21.4 | 0.41 |
| Adjusted PGA | Continuous | 326 | 1.09 | 0.47–2.53 | 0.84 |
| IDC-P/CA | | | | | |
| | Absent | 222 | – | – | – |
| | Present | 104 | 1.58 | 0.67 - 3.57 | 0.27 |

The prognostic impact of *ZNRF3* loss was evaluated using a multivariable Cox Proportional Hazards model, with pre-treatment PSA, Gleason Score, Clinical T-Category, adjusted PGA (i.e. PGA with chromosome 22 excluded), and IDC-P/CA as co-variates. *p*-values calculated from a Wald test. Source data are provided as a Source Data file.

cancers, we determined global RNA abundance patterns related to *ZNRF3*. To maximize the likelihood of identifying functionally important correlations, we looked for associations with *ZNRF3* loss in both the CPCG and TCGA cohorts. Using the Molecular Signatures Database (MSigDB; http://software.broadinstitute.org/gsea/msigdb/annotate.jsp), we identified 140 genes with a Spearman's $\rho \geq 0.4$ in both cohorts (Supplementary Data 5). Of these, five were located on chromosome 22, and were therefore excluded from the downstream analyses. The remaining 135 genes were enriched for genes downregulated in metastatic prostate cancer[37] (13/135; $q = 2.2 \times 10^{-7}$, Hypergeometric test) and for genes involved in positive regulation of canonical WNT signaling (GO: 0060070; 9/140, $q = 7.57 \times 10^{-4}$, Hypergeometric test); genes implicated in WNT signaling with significantly different RNA abundance in both CPCG and TCGA cases harboring *ZNRF3* loss are shown in Table S9. These data suggest that *ZNRF3* downregulation in localized prostate cancer may activate WNT signaling, an established driver of mCRPC[16,21].

**Multiple ZNRF3 features are associated with adverse outcomes in localized prostate cancer.** The vast majority of *ZNRF3* losses are monoallelic. In TCGA, for example, 7/65 (10.8%) *ZNRF3* losses are biallelic, while across both mCRPC cohorts, 7/157 (4.5%) *ZNRF3* losses are biallelic. As such, other mechanisms—including epigenetic silencing—may contribute to the downregulation of ZNRF3 RNA observed in aggressive localized disease. We therefore analyzed global DNA methylation patterns in localized prostate cancers. In both the CPCG[38] and TCGA cohorts, the CpG most significantly differentially methylated between patients with and without elevated *ZNRF3* RNA was located in the *ZNRF3* 5′ promoter region (probe ID: cg11986861; Fig. 4A and S11A, B). Methylation of this CpG was increased by 1.4-fold and 1.7-fold in CPCG and TCGA cases with low ZNRF3 RNA abundance, respectively, and was inversely correlated with ZNRF3 RNA abundance (CPCG: $\rho = -0.41$, $p = 3.19 \times 10^{-9}$; Fig. 4B; TCGA: $\rho = -0.50$, $p = 1.12 \times 10^{-31}$; Fig. S11C). Despite the loss of one *ZNRF3* allele, this CpG was also significantly hypermethylated in tumors with *ZNRF3* loss (CPCG: $p = 9.16 \times 10^{-3}$; TCGA: $p = 2.48 \times 10^{-3}$, Mann–Whitney U test; Fig. 4C, D). *ZNRF3* hypermethylation was associated with increased risk of metastatic relapse in CPCG (HR: 2.18, 95% CI: 1.06–4.05; $p = 3.47 \times 10^{-2}$; Wald test) and of shorter progression-free survival in TCGA (HR: 2.19, 95% CI:

1.45–3.31; $p = 1.83 \times 10^{-4}$; Wald test). ZNRF3 RNA abundance was significantly lower in tumors harboring one or more *ZNRF3* alteration (i.e. monoallelic loss, low RNA abundance, and/or hypermethylation; $p = 5.42 \times 10^{-34}$, one-way ANOVA; Fig. S11D) and those tumors harboring >1 alteration had significantly lower ZNRF3 RNA abundance than those harboring only a single alteration ($p = 1.78 \times 10^{-3}$, Tukey post-hoc test). In both the CPCG and TCGA cohorts, patients whose tumor genomes harbored more than one *ZNRF3*-associated feature were at significantly higher risk of adverse outcomes than those whose genomes harbored one or fewer (Fig. 4E, F).

**ZNRF3 loss is associated with activation of cell cycle progression and PRC1/2 pathways.** We next assessed associations between *ZNRF3* loss and clinical, pathological, and genomic features in the CPCG cohort (Fig. 5A). *ZNRF3* genomic loss was not associated with age at diagnosis ($q = 0.491$, Mann–Whitney U test). Moreover, *ZNRF3* genomic loss was not associated with the presence of IDC/CA, either across the full cohort ($q = 0.491$, Fisher's Exact test) or when patients were stratified by ISUP grade group (to account for the increased prevalence of IDC/CA in higher grade tumors[39,40]; ISUP grade group 1: $q > 0.99$; ISUP grade group 2: $q = 0.43$; ISUP grade groups 3 and 4: $q > 0.99$; Fisher's Exact tests). IDC/CA was prognostic of metastatic relapse on univariable analysis ($HR_{METS} = 2.14$, 95% CI: 1.12–4.52, $p = 0.046$, Wald test), as previously reported[41,42]. However, on multivariable analysis including *ZNRF3* loss, ISUP grade, PSA, clinical T-category, and adjusted PGA ($N_{METS} = 30/326$; Fig. S10), IDC/CA was not prognostic. *ZNRF3* loss was not significantly associated with ETS fusion status (Supplementary Data 6), chromothripsis ($p = 0.208$, Fisher's Exact test), kataegis ($q = 0.298$, Fisher's Exact test), global SNV burden ($p = 0.376$, Mann–Whitney U test), or tumor hypoxia, measured either using a consensus RNA abundance surrogate signature[43] ($p = 0.518$, Mann–Whitney U Test) or by direct intratumoral oxygen measurements[32,44] ($p = 0.550$, Mann–Whitney U Test). We observed a significant enrichment of CNAs in *APC* (loss) or *CTNNB1* (gain) in tumors harboring *ZNRF3* loss (APC Loss + *ZNRF3* Loss: OR = 2.08, $q = 1.06 \times 10^{-4}$, Fisher's Exact test; *CTNNB1* Gain + *ZNRF3* Loss: OR = 1.92, $q = 2.50 \times 10^{-3}$, Fisher's Exact test; Either CNA + *ZNRF3* Loss: OR = 2.24, $q = 9.81 \times 10^{-7}$, Fisher's Exact test). SNVs in either *APC* or *CTNNB1* were not associated with ZNRF3 loss (APC: $p = 0.843$;

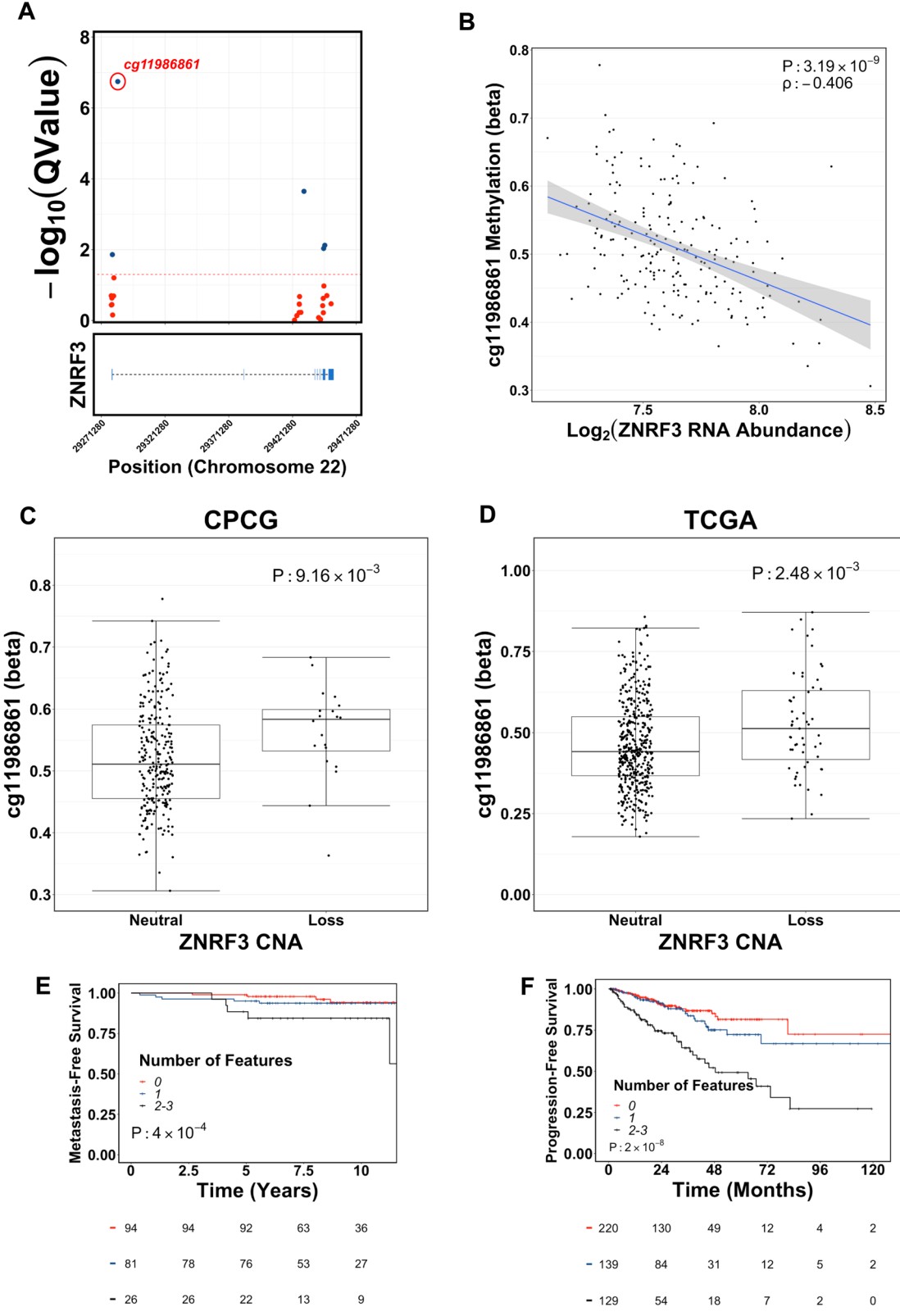

CTNNB1: $q = 1$, Fisher's Exact tests). On multivariable analysis, *ZNRF3* and *APC* loss were independently prognostic of PFS in TCGA ($N_{PFS} = 91$) and only *ZNRF3* loss was independently prognostic of metastatic relapse in CPCG ($N_{METS} = 36$; Table S10).

To assess the potential functional role of *ZNRF3* loss in localized prostate cancer, we performed Gene Set Enrichment Analysis (GSEA) in tumors with or without *ZNRF3* loss. We

initially focused on TCGA cases to capitalize on the larger number of samples with RNA abundance data ($n = 493$). GSEA of Gene Ontology (GO) pathways revealed enrichment of cell cycle progression gene sets in tumors harboring *ZNRF3* loss (Fig. 5B), which validated in two independent cohorts of both localized and metastatic prostate cancer (Fig. S12A, B). Given the enrichment of cell cycle progression gene sets in tumors

**Fig. 4 Aberrant methylation of ZNRF3 in aggressive localized prostate cancer. A** The *ZNRF3* 5′ promoter (probe cg11986861) is significantly hypermethylated in localized prostate cancers harboring low ZNRF3 RNA abundance. Coordinates refer to the GRCh37 (hg19) human genome build. Blue dots represent probes with statistically significant differential methylation (Q < 0.05). **B** ZNRF3 RNA abundance is inversely associated with methylation of the *ZNRF3* 5′ promoter. Shaded area shows 95% confidence interval of the best fit line. Two-sided *p*-value generated from a permutation test. **C, D** *ZNRF3* 5′ promoter methylation is significantly elevated in tumors harboring ZNRF3 allelic losses. *p*-values from a Mann–Whitney U test. CPCG: *n* = 286 patients; TCGA: *n* = 493 patients. Centre of box represents the median value. Lower and upper box hinges correspond to the first and third quartile. Whiskers extend to the largest and smallest values no further than 1.5-times the Interquartile Range. **E, F** Patients whose tumors harbor >1 aberrant ZNRF3-associated feature (i.e. monoallelic loss, low RNA abundance, 5′ promoter hypermethylation) are at significantly higher risk of metastatic relapse (**E**; CPCG) and disease progression (**F**; TCGA) than those patients with one or no features. *p*-values from a log-rank test. Colors indicate the number of features present in the tumor. Source data are provided as a Source Data file.

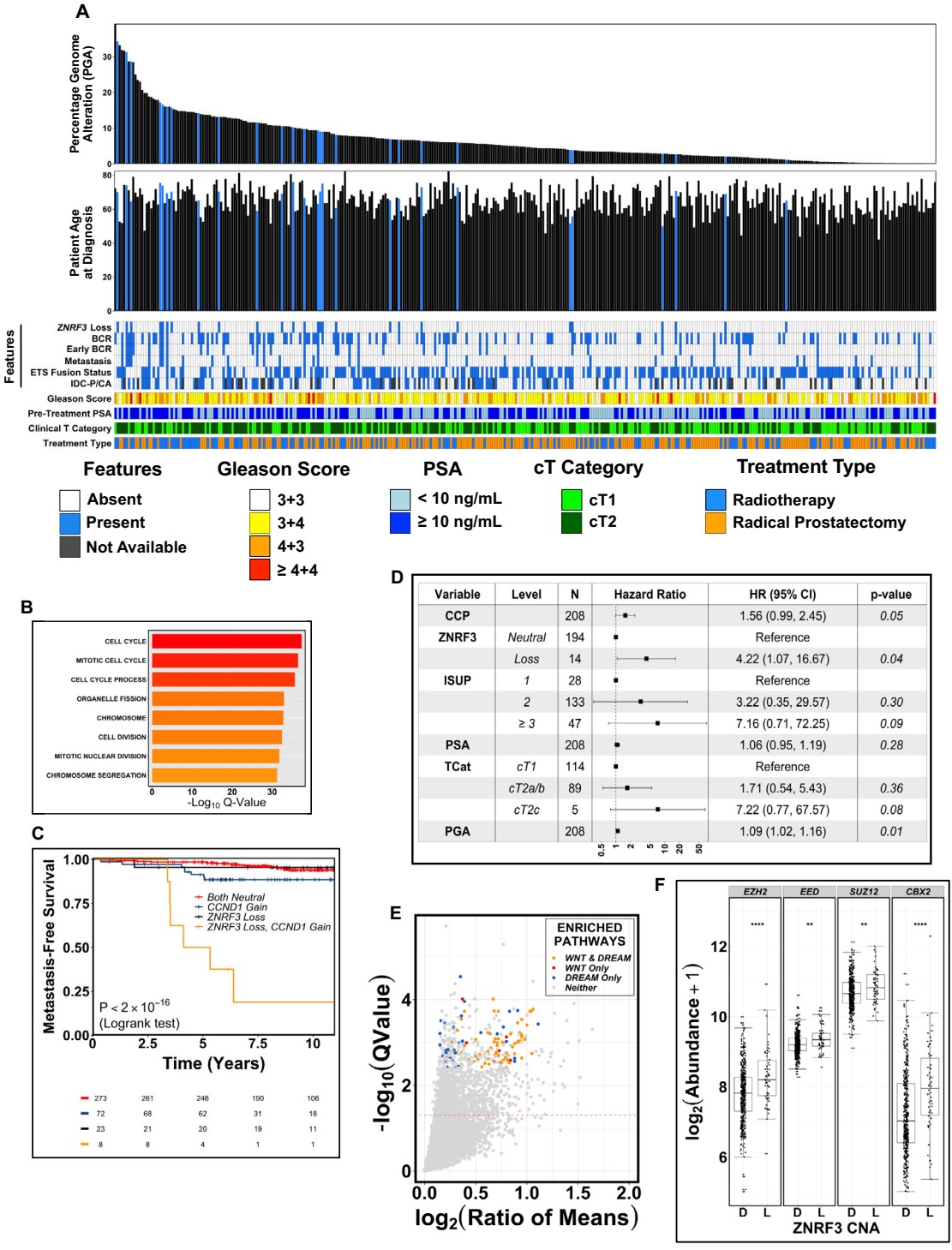

**Fig. 5 Molecular and clinical correlates of ZNRF3 genomic loss. A** Patients in the CPCG cohort ordered from left to right by percentage of the genome altered by a copy number aberration (PGA). Patient age at diagnosis (years) is also shown. Blue bars correspond with patients harboring *ZNRF3* genomic loss. Heatmaps below show clinico-molecular features (ZNRF3 Loss, BCR at any time, BCR within 30 months of treatment, Metastasis, ETS Fusion status, and IDC-P/CA histology; 'present' in blue, 'absent' in white, 'not available' in gray), clinical prognostic factors (Gleason Score, pre-treatment PSA, and clinical T category), and treatment type (image-guided radiotherapy; IGRT, blue or radical prostatectomy; RP, orange); **B** Gene Set Enrichment Analysis of TCGA tumors harboring *ZNRF3* loss. Colors represent *q*-value gradient. **C** Metastatic relapse-free rate in CPCG patients, stratified by *ZNRF3* genomic loss and *CCND1* genomic gain. *p*-value from a log-rank test ($p = 7.48 \times 10^{-23}$). Colors indicate different molecular profiles in each group. **D** Multivariable Cox Proportional Hazards analysis of metastatic relapse in CPCG patients, stratified by *ZNRF3* genomic loss, CCP/Prolaris score (continuous), clinical prognostic factors, and PGA. *p*-values from a Wald test. Error bars represent 95% confidence intervals of the reported hazard ratios. **E** Volcano plot of genes upregulated in TCGA tumors harboring monoallelic *ZNRF3* loss. Dashed line represents cutoff of statistical significance ($q > 0.05$). Colors show genes implicated in the DREAM (blue) or WNT (red) pathways, or both pathways (orange). **F** Significant RNA upregulation of genes implicated in polycomb repressive complex-1 and -2 (PRC2: *EZH2, EED, SUZ12*; PRC1: *CBX2*) in TCGA tumors ($n = 493$) harboring monoallelic losses of *ZNRF3*. \*\*\* $p < 0.0001$; \*\* $p < 0.001$. *p*-values from a Mann–Whitney U test. Centre of box represents the median value. Lower and upper box hinges correspond to the first and third quartile. Whiskers extend to the largest and smallest values no further than 1.5-times the Interquartile Range. Source data are provided as a Source Data file.

harboring *ZNRF3* loss, we evaluated clinical outcomes in patients with *ZNRF3* genomic loss, with or without amplification of *CCND1*. We focused on *CCND1* because its gain was significantly more prevalent in mCRPC than localized disease and was itself associated with metastatic relapse of localized disease; Fig. 2A and Table S1). While only eight patients harbored CNAs of both *ZNRF3* and *CCND1*, these patients were at significantly elevated risk of BCR and metastatic relapse ($p < 2 \times 10^{-16}$, log-rank test; Fig. 5C and S11C). To further confirm the effects of *ZNRF3* and cell proliferation, we assessed the interplay between *ZNRF3* loss and a clinically-validated RNA-based prognostic biomarker (Prolaris/CCP), which is based on abundance of 31 genes related to cell cycle progression[45]. CCP score predicted risk of metastasis (HR = 1.51, 95% CI: 0.985–2.33, $p = 0.059$, Wald test) in CPCG. However, on multivariable Cox proportional hazards analysis in CPCG, including CCP score, *ZNRF3* loss, PGA, and clinical prognostic factors, only *ZNRF3* loss and PGA remained prognostic of metastatic relapse at our pre-specified level of statistical significance ($p < 0.05$; $N_{\text{METS}} = 17/208$; Fig. 5D).

GSEA also revealed strong enrichment of genes implicated in dimerization partner, RB-like, E2F and multi-vulval class B (DREAM) complex signaling and WNT activation and in Polycomb Repressive Complex-1 and -2 (PRC1/2) signaling in tumors harboring ZNRF3 loss, including *EZH2, SUZ12, EED,* and *CBX2* (Fig. 5E, F). Upregulation of PRC1/2 is strongly linked to prostate cancer progression, lineage plasticity, and prostate cancer-specific mortality[46–50]. Taken together, these data demonstrate a link between *ZNRF3* loss and activation of pathways with well-established links to aggressive prostate cancer.

## Discussion

Prostate cancer lethality is a corollary of distant metastasis. While most newly diagnosed cases are localized with no evidence of extra-glandular spread, many men relapse following apparently successful local therapy. This implies the presence of occult metastases at the time of treatment, which are not accurately predicted using standard clinical prognostic factors. We and others have used multi-omic approaches to identify candidate biomarkers that predict BCR[7–10,51–53]. While BCR is an early and easily measured endpoint, it is a weak surrogate of prostate cancer-specific mortality[5,6]. Assessment of more meaningful clinical endpoints (e.g. metastasis) requires mature cohorts with matched molecular data and long-term follow-up. Moreover, because large multi-omic studies interrogate many thousands of individual mutations for links to outcome, there is a significant risk of false discovery due to multiple hypothesis testing.

To overcome these limitations, we assessed differential prevalence of a set of established prostate cancer driver and recurrent

mutations in localized and metastatic prostate cancer to identify candidate prognostic biomarkers of metastasis. In addition to validating established biomarkers of biochemical relapse—*MYC*, *CCND1*, and *PRKDC* gain, *TP53* loss—as also prognostic of metastatic relapse of localized disease, we identified and validated *ZNRF3* copy number loss as a prognostic biomarker of metastatic relapse of localized disease. Importantly, *ZNRF3* loss is associated with higher tumor grade but provides prognostic value independently of grade and other clinical prognostic factors. Of note, neither *PTEN* nor *RB1* losses were prognostic of metastatic relapse in CPCG, although both were prognostic of PFS in the TCGA cohort. This may be due to the reduced power of multi-gene studies (based on genome-wide surveys) such as this one relative to candidate gene analyses, which have previously suggested a role for these tumor suppressors in event-free survival and metastatic relapse[54]. Alternatively, our findings may, in part, reflect a unique biology associated with the more advanced disease enriched in the TCGA relative to the CPCG one.

WNT activation is a hallmark of mCRPC[21,22], and CNAs and hypomethylation of WNT pathway genes occur in localized prostate cancer in germline *BRCA2* mutation carriers[55,56], which have a significantly more aggressive clinical course than sporadic disease[57]. This implicates WNT signaling in the development of aggressive, potentially lethal prostate cancer, consistent with our finding—across multiple independent cohorts—of a link between *ZNRF3* loss, WNT pathway activation, and adverse clinical outcomes. Similarly, the strong upregulation of cell cycle progression, E2F/DREAM, and PRC1/2 pathways in tumors harboring *ZNRF3* loss further supports a link between this tumor suppressor and pathways with established links to prostate cancer aggression.

*ZNRF3* genomic loss has not been identified as prognostic for overall survival in mCRPC[22]. Allelic gains in *CTNNB1* (26% of mCRPC specimens) were not prognostic of shorter overall survival in mCRPC. These findings are consistent with a model in which WNT pathway activation—via *ZNRF3* loss, *CTNNB1* allelic gains, or other mechanisms—increases the risk of metastatic spread of the primary tumor but is not required for disease progression once metastases have formed. Our finding that *ZNRF3* loss or low ZNRF3 RNA abundance is associated with upregulation of cell cycle progression pathways further supports a model whereby WNT activation may promote aggressive disease in mitotically active cancers. While *ZNRF3* loss is comparatively rare in localized disease, the apparent interaction of at least two mechanisms of *ZNRF3* silencing (loss, hypermethylation) suggests that the detection of loss alone, may underestimate the impact of this gene as a predictor of adverse outcomes. However, longitudinal studies of patient-matched primary disease—ideally taking into account the presence of multifocal disease—and

distant metastases will be required to establish the precise mechanisms through which these (and other) aberrations contribute to aggressive prostate cancer.

These data require careful prospective validation and testing in randomized trials but suggest several potential clinical applications for *ZNRF3* loss. For example, patients with favorable intermediate risk prostate cancer whose tumors harbor *ZNRF3* loss might be triaged out of active surveillance protocols and toward definitive therapy. Similarly, patients with unfavorable intermediate risk disease who would otherwise receive curative-intent RP or radiation might be triaged to radiation + a short course of ADT, as is now standard of care for men with high-risk disease. *ZNRF3* loss has also been suggested as a potential predictive biomarker for sensitivity to porcupine inhibitors[58], which are in clinical trials for WNT- and NOTCH-driven cancers (e.g. NCT01351103)[59]. Thus, while the precise clinical impact of assessing *ZNRF3* loss is unclear, its presence in ~10% of localized disease could make it a valuable endpoint to routinely assess. This is particularly true in the context of more frequent somatic[60] or germline[25] molecular testing, where the incremental cost of assessing an additional gene would be low. Moreover, while both *ZNRF3* loss and ZNRF3 RNA downregulation are significantly associated with higher grade disease, these features conferred a risk of adverse outcomes independently of grade. This suggests that assessment of *ZNRF3* loss could help to identify a substantial fraction of patients who are at high risk of metastatic relapse, even amongst those with higher grade disease. Thus, in some sense the 10% overall rate of *ZNRF3* loss is misleading, because it encompasses all localized disease. Those 10% of cases are strongly enriched for men at the highest risk of metastatic relapse.

Multiple lines of evidence support the hypothesis that *ZNRF3* contributes to the clinical aggression observed in patients harboring monoallelic chr22q12.1 loss. Of the 9 genes in the region that were significantly associated with metastasis-free survival when downregulated at the RNA level, low ZNRF3 RNA abundance was the only one that was prognostic across four independent validation cohorts. Nevertheless, we cannot exclude the possibility that other genes in the chr22q12.1 region contribute to this aggressive phenotype. Moreover, the finding that tumors harboring >1 *ZNRF3* alterations (i.e. monoallelic loss, low RNA abundance, and/or 5' hypermethylation) are significantly more aggressive than those harboring 0–1 alterations supports a role for *ZNRF3* in promoting disease aggression. These data are consistent with the hypothesis that monoallelic loss of *ZNRF3* and epigenetic silencing of the remaining allele may each contribute to a reduction in ZNRF3 RNA abundance and increased tumor aggression.

The current study has some limitations; first, the microarrays used for calling CNAs in the CPCG and TCGA cohorts offer lower resolution than the sequencing-based approaches used in other cohorts. This may result in under-calling of focal CNAs leading to some driver CNAs that exhibit modestly increased prevalence in mCRPC being false positives. *ZNRF3* loss was 3-fold more prevalent in mCRPC than localized disease and was detected by microarray within a region of a minimum observed size of ~1 Mbp (median: 6 Mbp, range: 1.01–35 Mbp), though we cannot exclude the possibility that some cases may harbor focal CNAs in this region. Similarly, mutation calling based on whole-genome sequencing may have lower sensitivity than higher depth whole-exome sequencing, although the overall prevalence of SNVs in the driver genes studied was not different in whole-exome vs. whole-genome sequencing mCRPC studies (Whole Exome: 5.52%; 1078 SNVs in 454 samples × 43 genes assessed per sample vs. Whole Genome: 5.39%; 234 SNVs in 101 samples × 43 genes assessed per sample; $p = 0.284$, Pearson's $X^2$ test with Yates' correction). Third, the vast majority of localized prostate cancer

specimens were derived from surgically resected tissue while all mCRPC specimens were derived from biopsied metastases; the latter may have lower clonal complexity than surgical specimens. Thus, some of the mCRPC tumors may harbor subclonal mutations in driver genes which were not detected due to sampling bias. Fourth, driver SVs are under-represented across the study because several studies employed exome sequencing. Fifth, the cohorts in the study are strongly biased toward Caucasian men, and it is unclear to what extent these findings will generalize across ancestries. Sixth, the localized cohort is largely composed of intermediate risk cases. While these represent a plurality of newly diagnosed cases, it is unclear how these findings relate to other localized disease states. Finally, we cannot exclude the possibility of study-specific false-negative errors due to the use of unique analysis pipelines for each study cohort, although, the current study detected clinically relevant mutations in at least some driver genes across validated analysis pipelines[61–63]. For example, in localized disease, CNAs were called from either whole-genome sequencing (Gerhauser) or various microarray platforms (CPCG, TCGA, Baca, Barbieri, Taylor). *ZNRF3* loss frequencies were not significantly different across these cohorts (WGS: 5/48 vs. array: 108/1038; $p = 0.811$, Pearson's $X^2$ test with Yates' correction), as might be expected if detection sensitivity were inflating the CNA frequency in WGS-based studies.

Comparison of primary and metastatic tumor genomics provides an attractive strategy for prognostic biomarker discovery. We apply it to identify *ZNRF3* as a predictor of metastatic relapse in localized prostate cancer. Pre-treatment evaluation of *ZNRF3* tumor genomic loss and RNA abundance might improve treatment stratification for men with localized prostate cancer.

## Methods

**Patient cohorts, pathology, and tissue procurement**. Patients in the Canadian Prostate Cancer Genome Network (CPCG) cohort (Supplementary Data 1; $n = 385$) were consented for whole-genome sequencing and other molecular analyses and for reporting of anonymized clinical data, with approval from local Research Ethics Boards (UHN #11-0024 and 06-0822; CHUQ 2012-913:H12-03-192). All patients had National Comprehensive Cancer Network (NCCN) intermediate risk prostate cancer, were treated with radical prostatectomy (RP) or external-beam image-guided radiotherapy (IGRT) and were hormone- and chemotherapy-naïve at the time of treatment. Whole-blood or buffy coat specimens were acquired at the time of consenting. For patients undergoing RP, a fresh-frozen specimen was obtained from the index lesion within the resected prostate. For patients undergoing IGRT, an ultrasound-guided biopsy to the index lesion was obtained prior to the start of radiotherapy and was flash frozen in optimal cooling temperature (OCT) medium. For all specimens, 20 × 0 μm sections were acquired, with a hematoxylin and eosin (H&E)-stained 5 μm section on the top and bottom, as well as between the 10th and 11th section, to confirm continuity of histological features. All specimens were independently audited by two urogenital pathologists for Gleason/ISUP grade[7], tumor cellularity, and presence of intraductal carcinoma of the prostate (IDC-P) and cribriform architecture (CA) histology. Specimens of at least 70% tumor cellularity were used for molecular analyses. Genomic DNA was extracted using phenol:chloroform, as previously reported[7,10]. Double-stranded DNA quantity was assessed using a Qubit fluorometer and quality was assessed using a Nanodrop spectrophotometer and a BioRad Bioanalyzer, as previously reported[7]. All clinical, pathological, and molecular data for the CPCG cohort have been reported elsewhere[7,15,43,51,64].

Molecular, clinical, and pathologic data for patients in the TCGA PRAD cohort (PanCancer Atlas; $n = 494$; Supplementary Data 1)[19] were obtained from cBioPortal[65,66] (www.cbioportal.org) and the NIH Genomic Data Commons Data Portal (https://portal.gdc.cancer.gov). Similar data were obtained from the publicly available Baca[11], Berger[12], Weischenfeldt[13], Barbieri[20], Gerhauser[14], Robinson[21], and Abida[22] cohorts (Supplementary Data 1). Where an individual patient sample was included in multiple reports (e.g. samples from the Baca, Berger, Barbieri, Weischenfeldt, and TCGA studies were included in a meta-analysis along with CPCG samples[7], the Robinson/Abida SU2C mCRPC cohorts[21,22] contained 125 patients that were analyzed in both studies), all cases were audited using original sample and patient identifiers across studies to ensure no duplication of patients. Data from the Quigley cohort[18] was downloaded from the authors' website, as described below. Data for the Taylor/MSKCC validation cohort[27] was obtained from cBioPortal.

Patients in the Mt Sinai Hospital cohort ('Mt Sinai'; $n = 47$) were consented for molecular analysis and reporting of anonymized clinical data, with approval from

the Research Ethics Board at Mt. Sinai Hospital and the Lunenfeld Research Institute (MSH REB #14-0211-E and University of Toronto REB #35275). All patients underwent RP for localized high-volume intermediate risk or high-risk prostate cancer and total RNA was extracted from fresh-frozen specimens (see below).

The final cohort for molecular discovery consisted of 1844 unique patient samples from the Abida, Baca, Barbieri, Berger, CPCG, Gerhauser, Quigley, Robinson, TCGA, and Weischenfeldt studies (Supplemenary Data 1). For clinical outcome analyses (see below), CPCG was used for discovery, with validation in the Gerhauser (EOPC), Mt. Sinai (LTRI), Taylor, and TCGA cohorts.

**Selection of driver aberrations**. A list of 113 mutations across 72 established driver genes or recurrently altered loci was compiled as the union of drivers reported in the Quigley and Fraser studies[7,18] (Supplementary Data 2). Mutations included copy number aberrations (CNAs), coding single nucleotide variants (SNVs; non-synonymous, stop codon gained, stop codon lost, and splice gain/loss), non-coding single nucleotide variants (ncSNVs) and non-CNA structural variants (translocations, fusions, inversions; SVs). While frameshift indels can also result in stop codon losses, we did not include these mutations in the current analysis, which focused only on established driver mutations. Mutations in the androgen receptor enhancer (624 kb upstream of $AR$[18]; CNAs only) were pooled with those in the $AR$ gene body. For ncSNV drivers identified in CPCG[7], hg19 coordinates were mapped to GRCh38 using the NCBI Remap web interface (https://www.ncbi.nlm.nih.gov/genome/tools/remap) to compare across cohorts. In cases where a gene was subject to multiple mutation types (e.g. $TP53$ SNVs, CNAs, and SVs), each mutation was analyzed independently. Because not all mutation types were available for all cohorts, the denominator was different for different types:

Coding SNV: 1204 localized + 555 mCRPC = 1759 specimens
Non-Coding SNV: 201 localized + 101 mCRPC = 302 specimens
CNA: 1279 localized + 555 mCRPC = 1834 specimens
SV: 201 localized + 101 mCRPC = 302 specimens

**DNA copy number aberrations**. For the CPCG cohort, copy number aberrations (CNAs) were called from OncoScan FFPE v3 microarrays ($n = 382$), as previously described[7]. For The Cancer Genome Atlas Prostate Adenocarcinoma (TCGA-PRAD; $n = 494$) and Baca ($n = 53$) cohorts, CNAs were called from Illumina SNP 6.0 microarrays. For the Taylor/MSKCC validation cohort ($n = 194$), CNAs were called from Agilent 244 K array comparative genomic hybridization (aCGH) microarrays. Overall, CNA calls were available for 1279/1289 localized patients. For mCRPC specimens from the Quigley study ($n = 101$), CNAs were called from whole-genome sequencing ($n = 101$) using CopyCat-generated BED files downloaded from the authors' website (http://davidquigley.com/prostate.html) with copy number status ('gain', 'neutral', or 'loss') assigned based on $\log_2$ ratios according to the authors' guidelines[18]: for autosomal genes, gain: $\log_2$ CN score ≥ 3, loss: $\log_2$ CN score ≤ 1.65, neutral: 1.65 < $\log_2$ CN score < 3; for genes on chromosomes X and Y, gain: $\log_2$ CN score ≥ 1.4; loss: $\log_2$ CN score ≤ 0.6; neutral: 0.6 < $\log_2$ CN score < 1.4. For the Abida and Robinson mCRPC cohorts ($n = 454$), CNAs were called from whole-exome sequencing, as described[21]. In all cases except CPCG and Quigley cohorts, CNA data were extracted from publicly available datasets via the CGDS-R package (v 1.3.0). In these cases, shallow deletion and deep deletion were pooled as loss while gain and amplification were pooled as gain.

Percentage of the genome affected by a copy number alteration (PGA) was calculated as the number of bases affected by a CNA divided by the total number of bases in the genome, as previously reported[7,32,67,68]. When comparing the PGA associated with the presence or absence of a specific CNA, an adjusted PGA was calculated by omitting the chromosome on which the specific gene is found.

**Single nucleotide variants**. Coding single nucleotide variants (SNVs) in driver genes were called from whole-genome or whole-exome sequencing data based on tumor-normal comparisons. Of the 1289 localized prostate cancer patients, coding SNV data were available from 1204 independent localized patient specimens (CPCG: 300, Barbieri: 109, Berger: 7, Baca: 53, TCGA: 494, Weischenfeldt: 11, and Gerhauser: 230) and 555 mCRPC patient specimens (Quigley: 101, SU2C: 454). As noted above, these numbers represent unique patients; where a given specimen was included in two or more studies, it was included only once in the current study. SNV data for all localized prostate cancer studies were downloaded from cBio-Portal (via the CGDS-R package for R) or the ICGC Data Portal (dcc.icgc.org). For Quigley mCRPC specimens, VCF files were downloaded from the authors' website (http://davidquigley.com/prostate.html). For Robinson and Abida cohort mCRPC specimens, SNV calls were downloaded into R from cBioPortal using the CGDS-R package (v1.3.0). For each patient, we extracted calls for missense (non-synonymous), nonsense (stop codon gained or stop codon lost), and splicing variants (spice donor or splice acceptor) within each gene analyzed.

For non-coding SNVs in the Quigley cohort, hg19 coordinates from the CPCG study were re-mapped to GRCh38 (as described above).

The list of genes affected by SNVs in the current study is available in Supplementary Data 2. Amongst the 43 genes evaluated for SNVs, 20 were included in a previous analysis of driver SNV enrichment in localized prostate cancer vs. mCRPC[16].

**Structural variants**. Driver structural variants (SVs) in the CPCG cohort were previously reported[7]. For mCRPC specimens, SV calls from Manta[69] were downloaded from the authors' website (http://davidquigley.com/prostate.html). Overall, SV calls were available for 201 localized patients and 101 mCRPC patients. SVs included translocations and inversions, except where a specific SV type is specified for a given gene or locus. To assess SVs that were originally called from localized specimens in one megabase bins based on hg19 coordinates, these bins were re-mapped to GRCh38 using the NCBI Genome Remapping Service (https://www.ncbi.nlm.nih.gov/genome/tools/remap). For SVs at the $PTEN$ locus, we evaluated inter-chromosomal translocations and deletions separately from inversions within the chr10:89 Mbp bin (hg19), which regulate PTEN RNA abundance in localized prostate cancer[7].

**RNA abundance data**. RNA abundance data were available for 208 patients with clinical outcome data in the CPCG cohort[51]. For all cases, total RNA was extracted from tumor tissue sections, alternating with those used for whole-genome sequencing to minimize any effects of spatial heterogeneity. Total RNA (100 ng) was assayed using Affymetrix Human Transcriptome Array 2.0 and HuGene 2.0 ST microarrays, and RNA abundance calculated as previously reported[7,10]. Samples were stratified as having high or low RNA abundance based on median dichotomization of $\log_2$ abundance values.

The LHRI cohort RNA abundance was assessed from RNA-seq ($n = 47$). Briefly, total RNA was extracted using RNeasy Mini kit (Qiagen, Maryland, USA). RNA quantity was assessed using a Qubit fluorometer (ThermoFisher, Massachusetts, USA). 200 ng of total RNA was used to construct a TruSeq strand specific library with the Ribo-Zero protocol (Illumina), and all samples were sequenced on a HiSeq2500 to a target depth of 50 million read pairs. Reads were mapped and mRNA abundance was quantified using the STAR aligner (v2.5.2a) against GRCh37 with Gencode Annotations (v24) lifted to GRCh37[70]. Library normalization was performed using trimmed means of M-values (TMM) with the BioConductor package EdgeR (v3.12.1)[71]. Samples were stratified as having high or low RNA abundance based on median dichotomization of $\log_2$ abundance.

RNA abundance data for TCGA (RNA-seq; $n = 493$) and Taylor/MSKCC (Affymetrix Human Exon 1.0 ST microarrays; $n = 216$) were downloaded into R from cBioPortal (www.cBioPortal.org) using CGDS-R (v1.3.0). To take advantage of the increased statistical power from these larger cohorts, we stratified samples into quartiles based on $\log_2$ RNA abundance values.

Comparison of RNA abundance between groups was performed using the Spearman rank correlation coefficients and Mann–Whitney U tests. Gene Set Enrichment Analysis (GSEA) and Gene Ontology (GO) analyses were performed using the online Molecular Signatures Database tool from the Broad Institute (v7.0; March 3, 2020; http://software.broadinstitute.org/gsea/msigdb/annotate.jsp) an G:Profiler g:GOSt (version e99_eg46_p14_f929183d, July 20, 2020; https://biit.cs.ut.ee/gprofiler/).

**Comparison of driver aberration prevalence**. For the localized and mCRPC cohorts (as well as the two cohorts combined), we calculated the proportion of samples harboring mutations in each driver gene (e.g. $TP53$) as well as the proportion of samples harboring each driver gene mutation type (e.g. $TP53$ SNVs, CNAs, and SVs). For each individual mutation class (CNA, SNV, or SV), the proportion ($P_{MUT}$) was calculated as

$$P_{MUT} = \frac{\text{number of specimens with specific mutation}}{\text{number of specimens analysed for mutation class}} \quad (1)$$

We also calculated the proportion of specimens harboring more than one mutation class (i.e. CNA + SNV, CNA + SV, SNV + SV, and/or CNA + SNV + SV). 95% confidence intervals for the proportions of each mutational class (or combination of classes) were calculated as:

$$CI = p \pm 1.96\sqrt{p(1-p)/n} \quad (2)$$

where $p$ is the overall proportion for each mutation or combination of mutations for each gene and $n$ is the number of samples analyzed for that mutation class or combination of classes.

For each gene, the final proportion of cases harboring a mutation of any kind ($P_{GENE}$) was calculated as

$$P_{GENE} = (P_{CNA} + mP_{SNV} + P_{SV}) - (P_{CNA\&SNV} + P_{CNA\&SV} + P_{SNV\&SV} + P_{CNA\&SNV\&SV}) \quad (3)$$

where $P$ is the proportion of cases harboring each stated mutation class (i.e. CNA, SNV, SV; as applicable to the specific gene). To account for unequal sample sizes for each mutation class (or combination of classes), we calculated propagation of error for each mutation class (or combination of classes, as applicable) as

$$PE_{class} = \sqrt{CI_{Upper}^2 + CI_{Lower}^2} \quad (4)$$

Variance of propagation of error was calculated as

$$\partial_{GENE}^2 = \sqrt{(PE_{CNA} + PE_{SNV} + PE_{SV}) - (PE_{CNA\&SNV} + PE_{CNA\&SV} + PE_{SNV\&SV} + PE_{CNA\&SNV\&SV})} \quad (5)$$

Final 95% confidence intervals for each gene were calculated as above:

$$CI = p \pm 1.96\sqrt{p(1-p)/n} \qquad (6)$$

where $p$ is the overall gene proportion and $n$ is the number of samples analyzed for that gene (Supplementary Data 2).

The per patient overlap in available mutation data is shown in Fig. S1. Differences in proportion of localized and mCRPC patients ('Δ proportion') were calculated by subtracting the proportion of localized specimens harboring the specific aberration from the proportion of mCRPC specimens harboring the same aberration. Positive Δ proportion values indicate higher prevalence in mCRPC; negative Δ proportion values indicate enrichment in localized disease. To determine if specific mutations were statistically differentially prevalent between localized disease and mCRPC, we used Fisher's Exact tests. For SNVs (coding and non-coding), we corrected for global mutational burden as follows: first, we used a binomial distribution to calculate the probability of observing no mutations in a given driver gene in a given sample, corrected for the size of the gene coding region (Mbp) and the mutational burden (SNVs per Mbp) in that sample ($P_{NOMUT}$); the probability of observing *at least* one mutation in a given gene was then calculated as: $P_{MUT} = 1 - P_{NOMUT}$. We then performed a simulation of 100,000 samples of the binomial distribution, using $P_{MUT}$ to weight the probability of mutation in each sample. A two-sided $p$-value was calculated as proportion of these simulations showing as great or greater simulated Δ proportion than observed Δ proportion.

We corrected for CNA burden in a similar manner, using gene size and per sample CNA burden (i.e. proportion of the genome altered by a CNA × 3 × 10⁹) to identify $P_{MUT}$ (as above), which was then used to weight the sampling probability. We then calculated a two-sided $p$-value for each driver CNA as half of the proportion of permutations showing as great or greater simulated Δ proportion than the observed Δ proportion (to account for the fact that a CNA can be either a gain or a loss).

Confidence intervals for Δ proportions were calculated using Yates' $X^2$ with continuity correction,

$$CI = \hat{p}_1 - \hat{p}_2 \pm 1.96\sqrt{\frac{\hat{p}_1(1-\hat{p}_1)}{m} + \frac{\hat{p}_2(1-\hat{p}_2)}{n}} \pm \frac{1}{2} * \left(\frac{1}{m} + \frac{1}{n}\right) \qquad (7)$$

where $p_1$ is the proportion of mCRPC harboring the mutation, $p_2$ is the proportion in localized prostate cancer harboring the mutation, $m$ is the number of mCRPC specimens tested, $n$ is the number of localized prostate cancer specimens tested.

Adjusted Δ proportion was calculated as the difference between observed Δ proportion and expected Δ proportion; adjusted Δ proportion values >0 indicate a higher-than-expected proportion of mCRPC cases harboring the specific mutation while adjusted Δ proportion <0 indicates a higher-than-expected proportion in localized disease.

**Clinical outcome**. For patients undergoing RP, biochemical relapse (BCR) was defined according to American Urological Association (AUA) guidelines: two consecutive PSA values of >0.2 ng/mL over at least 6 months following treatment or initiation of salvage therapy. Patients who had initial PSA failure after RP and then underwent successful salvage RT (i.e. PSA < 0.2 ng/mL in two consecutive tests within 6 months) were not classified as having BCR *unless* they subsequently met AUA (not Phoenix) conditions for BCR; in these cases, BCR was backdated to the time of initial post-RP PSA rise. For patients who underwent IGRT with curative intent, BCR was defined according to the Phoenix criteria[72]: a PSA value of 2 ng/mL above PSA nadir or initiation of salvage hormone therapy. For the TCGA cohort, progression-free, disease-specific, and overall survival were used as reported by the consortium[73].

Biochemical relapse-free rate (bRFR), metastatic relapse-free rate (mRFR) and overall survival were calculated using the Kaplan-Meier method. Associations between mutations and outcome were assessed using log-rank or univariate Cox proportional hazards models, as appropriate. Adjustments for clinical factors [T-category (categorical; T1, T2a/b, or T2c), pre-treatment PSA, and diagnostic (i.e. biopsy) ISUP grade (categorical; Grade 1 vs. Grade 2 vs. ≥ Grade 3)] using multivariable Cox proportional hazards modeling. In all cases, the proportional hazards assumptions were verified graphically using Schoenfeld residuals. Log-rank tests were used when the proportional hazards assumptions were violated.

**Prognostic signature scores**. Fraser signature scores were calculated based on a modified version of the six-feature signature reported in Fraser et al.[7]. Briefly, *MYC* gain, *ATM* SNVs, *TCERGL1* hypomethylation, *ACTL6B* hypermethylation, chr7:61 Mbp inter-chromosomal translocations, and clinical T category were scored for each patient; CNAs, SNVs, and SVs were scored as absent (0) or present (1); probe-based methylation β-values were median dichotomized and patients scored as being either above or below the median; clinical T category was scored as cT1 or cT2a/b (0) or cT2c (1). For multivariable analyses, a signature score was derived based on the sum of the six features. For visualization, scores were median dichotomized and patients assigned to either "Signature High" or "Signature Low" bins.

Cell Cycle Progression/Prolaris scores were approximated as previously reported[45,74]. Briefly, the mean abundance of the 31 CCP genes was normalized to the mean abundance of 15 housekeeping genes to yield a CCP score, as previously reported[45]. Outcomes analyses were performed using CCP as a continuous variable.

**Differential DNA methylation analyses**. Illumina HumanMethylation 450k microarray data for CPC-GENE were pre-processed as previously described[7]. TCGA PRAD 450k data were obtained from the TCGA data portal (https://portal.gdc.cancer.gov/projects/TCGA-PRAD). Beta values (i.e. ratio of methylated to unmethylated signal) were transformed to M-values (i.e. log₂ ratio of methylated probes to unmethylated probes) using the beta2m function within the lumi R package (v. 2.24.0)[75]. For each cohort, patients were classified as having high or low ZNRF3 RNA abundance, as described above. Differential methylation was assessed using the limma R package (v3.13)[76], including array probes mapping within genes ($n = 335,926$). Differentially methylated probes were visualized using Manhattan plots, with statistical significance inferred using a Bonferroni cutoff of $p < 1.49 \times 10^{-7}$. To test the association between methylation and clinical outcomes, $M$-values for each patient were dichotomized as high or low based on Youden's J statistic, using the cutpointr R package (v1.1.1)[77]. Patients were then binned based on this cutoff and time-to-event (metastasis or disease progression) was determined using Cox proportional hazards models, as above.

**Statistical testing and data visualization**. All statistical analyses were performed using R statistical software (v3.5.2) with the following packages: BoutrosLab.plotting.general (v5.9.2)[78], VennDiagram (v1.6.20)[79], CGDS-R (v1.3.0), survival (v2.43-3), ggplot2 (v3.1.0), easyGgplot2 (v1.0), survminer (v0.4.3), forestplot (v1.7.2), factorial2x2 (v0.2.0), cowplot (v1.1.0) and tidyverse (v1.2.1). All tests of statistical significance were two-sided. $p$-values were corrected for multiple comparisons using the Benjamini–Hochberg False Discovery Rate (FDR) method or the Bonferroni method, as noted. For Analysis of Variance (ANOVA), between group significance was evaluated using Tukey post-hoc tests. All source code is available from Zenodo[80].

**Reporting summary**. Further information on research design is available in the Nature Research Reporting Summary linked to this article.

## Data availability

The genomic variant calls, clinical, and pathology data for the CPC-GENE study are available through the ICGC Data Portal under accession PRAD-CA. Raw whole-genome sequencing data for the CPC-GENE study are available through the European Genome-Phenome Archive used accession ID EGAD00001003761. Raw DNA methylation data are available from the NCBI Gene Expression Omnibus under accession number GSE84043. Summary DNA methylation data are available from Figshare (https://doi.org/10.6084/m9.figshare.16574486.v1). Raw and processed mRNA array data for the CPC-GENE study are available from the NCBI Expression Omnibus under accession number GSE107299. Genomic variant calls, mRNA abundance, clinical, and pathology data are available from cBioPortal for the Abida (accession prad_su2c_2019), Barbieri (accession prad_broad), Baca (accession prad_broad_2013), Robinson (accession prad_su2c_2015), Taylor (accession prad_mskcc), Gerhauser and Weischenfeldt studies (accession prostate_dkfz_2018). Raw sequencing data, genomic variant calls, mRNA abundance, and DNA methylation data for the TCGA study are available from cBioPortal (accession prad_tcga_pan_can_atlas_2018) and the GDC Data Portal (accession TCGA-PRAD). Processed genomic variant, mRNA abundance, and clinical/pathology data for the Quigley study are available from the authors' website (https://quigleylab.ucsf.edu/data). Processed mRNA abundance, and clinical/pathology data for the LTRI cohort are available from Zenodo (https://doi.org/10.5281/zenodo.5389194). Patients in the LTRI cohort were not specifically consented for deposition of their raw sequencing data into public repositories. Therefore, raw RNAseq data for the LTRI cohort are available upon request from Dr. Michael Fraser (michael.fraser@cpcgene.com) and are subject to the requestor entering into a Data Sharing Agreement with the Lunenfeld-Tanenbaum Research Institute. We will attempt to make data available within one month of any request. Source data are provided as a Source Data file. Source data are provided with this paper.

## Code availability

Scripts used for data analysis and visualization are available from Zenodo (https://doi.org/10.5281/zenodo.5389194) and GitHub (https://github.com/mfraser3/ZNRF3_2021).

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

## Acknowledgements

Thanks to Drs. Kathleen Houlahan, Viniyak Bhandari, and John Watson for assistance with experimental design and statistical analysis and to all members of the Boutros lab for suggestions and support. This study was conducted with the support of Movember funds through Prostate Cancer Canada, and with the additional support of the Ontario Institute for Cancer Research, funded by the Government of Ontario. This work was supported by funds from the Department of Surgical Oncology and the Genetics and Epigenetics program, Princess Margaret Cancer Centre and the University of Toronto, Department of Surgery (Division of Urology) to Michael Fraser. Hansen H. He is supported by a Project Grant from the CIHR. This work was supported by Prostate Cancer Canada and is proudly funded by the Movember Foundation Team Grant T2013 and #RS2014-01 and a Prostate Cancer Canada Movember Discovery Grant (D2014-26) to Alexandre R. Zlotta. Paul C. Boutros was supported by a Terry Fox Research Institute New Investigator Award, a Prostate Cancer Canada Rising Star Fellowship, a CIHR New Investigator Award, a CIHR Project Grant, the Government of Canada through Genome Canada and the Ontario Genomics Institute (OGI-125), Canadian Cancer Society (grant #705649) and by the University of California. The authors gratefully thank the Princess Margaret Cancer Centre Foundation and Radiation Medicine Programme Academic Enrichment Fund for support (to Robert G. Bristow). This work was supported by a Terry Fox Research Institute Programme Project Grant. Robert G. Bristow is a recipient of a Canadian Cancer Society Research Scientist Award. Laboratory work for R.G.B. is supported by the CRUK Manchester Institute through Cancer Research UK. This work was supported by the NIH/NCI through awards P30CA016042, U01CA214194, and U24CA248265.

## Author contributions

Initiated the study: M.F., R.G.B., P.C.B. Pathology audits: T.v.d.K. Patient accrual: A.F., A.R.Z., R.G.B. Data analysis: M.F., J.L., J.W. Data visualization: M.F., P.C.B. Supervised study: M.F., H.H.H., J.W., T.v.d.K., A.R.Z., R.G.B., P.C.B. Wrote the first draft of the manuscript: M.F. Approved the final manuscript: all authors.

## Competing interests

The authors declare no competing interests.
