## [Peer review file · Nature Communications]

REVIEWER COMMENTS

Reviewer #1 (Remarks to the Author):

The authors have provided a thoughtful and detailed description of changes made to the manuscript based on the initial review. I believe that these changes have improved what was already an strong paper. I have no further comments.

Reviewer #2 (Remarks to the Author):

I feel the authors have done an excellent job responding to the reviewers' critiques, including additional data analyses and better addressing some of the limitations. I have no additional comments.

Reviewer #3 (Remarks to the Author):

Manuscript Title: Somatic Driver Mutation Prevalence in 1,844 Prostate Cancers Identifies ZNRF3 Loss as a Predictor of Metastatic Relapse

Authors: Fraser M et al

Summary: This is a revised manuscript that is focused on identifying molecular alterations that could serve as prognostic biomarkers in prostate cancer. The overall concept is to compare genomic alterations that occur more frequently in metastatic CRPC compared to localized PC. Existing datasets were used for comparisons of the prevalence of 'established' somatic driver SNVs, CNAs and SVs. The analyses culminated in the identification of ZNRF3, a WNT pathway family member. Associations between ZNRF3 monoallelic loss and various outcome parameters were shown.

Comments:

1. The authors provide a thoughtful point-by-point rebuttal/response to the prior critique and resolve a few of the minor comments.
2. Overall, this report continues to lack substantial novelty or innovation. As noted, the features selected for analyses were all previously identified in published studies – the 113 mutation types. The basic approach is not particularly new – though it is statistically rigorous – that is, comparing the frequency/prevalence of a given event in advanced versus localized cancer. The results are largely known: e.g AR, BRCA2, TP53 in metastatic cancers and SPOP more prevalent in localized cancers, etc.
3. No new patient/tumor-level data are provided.
4. The majority of the manuscript is focused on one finding – the association of ZNRF3 with advanced PC and the potential utility as a prognostic biomarker. However, if ZNRF3 is the major output for this study, it is underwhelming for the following reasons:
 - (i) ZNRF3 alterations are not common in localized prostate cancers – and thus will not be particularly useful for the vast majority of PC patients.
 - (ii) Though requested, no functional data regarding the role of ZNRF3 in any aspect of aggressive/metastatic behavior are provided.
 - (iii) As clarified in the revised manuscript and responses to the prior critique, there are 417 co-deleted with ZNRF3 on chromosome 22 and fully 29 of these were associated with biochemical relapse. Consequently – confidence that ZNRF3 independently drives adverse outcomes is not compelling and without confirmatory mechanistic data the conclusion is not well-supported.
 - (iv) The near-exclusive monoallelic copy loss and lack of any deleterious SNVs in ZNRF3 also suggests that ZNRF3 copy loss may not be the exclusive explanation for adverse outcomes in the setting of numerous other genomic alterations concurrent with ZNRF3 loss.

(v) ZNRF3 is not associated with adverse outcomes in all of the individual datasets comprising this analysis. The response is that ZNRF3 cooperates with other prognostic factors but this statement is not supported by strong mechanistic rationale.

(vi) As noted in the prior review, there are very few cases with biallelic loss of ZNRF3 and the authors appeared to suggest that ZNRF3 is functioning as a haplo-insufficient tumor/metastasis suppressor. The authors now agree with this possibility. However, no data are shown that clearly confirms this hypothesis.

5. The analyses failed to identify several gene/genomic alterations that have been clearly associated with adverse outcomes in many prior studies such as PTEN loss and state that the issue may be with the "reduced power of genome-wide studies such as this one relative to candidate gene analyses..". However, it is notable that the present study is not 'genome-wide' it is an analysis of 113 preselected genomic alterations.

6. Minor point – the authors state that ZNRF3 could serve as a predictive biomarker for porcupine inhibitors. However, it is not clear that any porcupine inhibitors are approved for any indication.

7. Overall, the impact of this study would be substantially increased if the ongoing preclinical studies alluded to in the authors responses involving prostate cancers with/without ZNRF3 loss demonstrated features associated with aggressive/metastatic behaviors and/or differential responses to WNT pathway antagonists.

Reviewer #3 (Remarks to the Author):

Manuscript Title: Somatic Driver Mutation Prevalence in 1,844 Prostate Cancers Identifies ZNRF3 Loss as a Predictor of Metastatic Relapse

Authors: Fraser M et al

Summary: This is a revised manuscript that is focused on identifying molecular alterations that could serve as prognostic biomarkers in prostate cancer. The overall concept is to compare genomic alterations that occur more frequently in metastatic CRPC compared to localized PC. Existing datasets were used for comparisons of the prevalence of ‘established’ somatic driver SNVs, CNAs and SVs. The analyses culminated in the identification of ZNRF3, a WNT pathway family member. Associations between ZNRF3 monoallelic loss and various outcome parameters were shown.

Comments:

1. The authors provide a thoughtful point-by-point rebuttal/response to the prior critique and resolve a few of the minor comments.

We sincerely thank the reviewer for their comments and careful critique of our work.

2. Overall, this report continues to lack substantial novelty or innovation. As noted, the features selected for analyses were all previously identified in published studies – the 113 mutation types. The basic approach is not particularly new – though it is statistically rigorous – that is, comparing the frequency/prevalence of a given event in advanced versus localized cancer. The results are largely known: e.g AR, BRCA2, TP53 in metastatic cancers and SPOP more prevalent in localized cancers, etc.

We agree with the reviewer that some of the mutations identified in our study have previously been shown to be differentially prevalent in mCRPC vs. localized disease. However, to our knowledge, there are no outcome-driven analyses of these differential mutation proportions. For example, the largest mCRPC/localized study in the literature to date (Armenia *et al*, *Nature Genetics*, 2018; PMID: 29610475) did not address the association between any differentially mutated gene and clinical outcomes in localized disease and focused on only exome sequencing therefore ignoring many critical classes of mutations, such as the genomic rearrangements widespread in prostate cancer. Given that the current manuscript analyses 831 more cases than Armenia *et al.*, fills in these “missing” mutation types and most importantly directly links specific differentially prevalent mutations to a clinically validated surrogate of prostate cancer-specific mortality (*i.e.* metastatic relapse), we feel that our manuscript adds significantly to the growing literature surrounding differentially mutated genes in localized prostate cancer vs. mCRPC.

We further note the vast majority of driver mutations found at higher prevalence in mCRPC – in both the current study and in Armenia *et al.* – did not predict metastatic relapse, including all of the mutations mentioned by the reviewer above. For example, of the 24 mutations found to be more prevalent in mCRPC and present in at least 5% of localized cancers (**Table S4**), only four

were significantly associated with metastasis-free survival. Thus, it does not necessarily follow that the presence, in localized prostate cancer, of any mutation that is more prevalent in mCRPC will predict aggressive localized disease.

3. No new patient/tumor-level data are provided.

The reviewer is correct. This is, to our knowledge, the largest outcome-linked analysis of publicly available prostate cancer genomics datasets to date, but does not include new genome sequencing data.

4. The majority of the manuscript is focused on one finding – the association of ZNRF3 with advanced PC and the potential utility as a prognostic biomarker. However, if ZNRF3 is the major output for this study, it is underwhelming for the following reasons:

(i) ZNRF3 alterations are not common in localized prostate cancers – and thus will not be particularly useful for the vast majority of PC patients.

While the reviewer is correct that ~10% of localized tumours harbour *ZNRF3* loss, 25-30% of cases that relapse metastatically after curative-intent local therapy have *ZNRF3* loss (**Supplementary Figure 5**), suggesting that a significant proportion of aggressive cases harbour this CNA. Moreover, this percentage is highly variable across patients with localized disease. For example, while **Supplemental Figure 6** shows that *ZNRF3* RNA downregulation is associated with higher grade disease, the same is true for *ZNRF3* loss. For example, of the 288 TCGA patients with GS 6 or 7 tumours, only 17 harboured *ZNRF3* loss (5.9%), while of the 201 patients with GS 8-10 tumours, 41 harboured *ZNRF3* loss (20.4%; OR = 0.245, $p = 1.28 \times 10^{-6}$, Fisher's Exact test). This is of clinical importance because *ZNRF3* loss predicts shorter time to disease progression and first metastatic relapse independently of tumour grade (**Figure 4D & Supplementary Figures 7 and 10**). We have added this new analysis to the revised manuscript, which now reads (lines 430-431):

“ZNRF3 loss was also associated with higher grade tumours in both CPCG and TCGA (Figure S4C).”

Fully appreciating the reviewer's point, we have also added the following to the discussion (lines 653-656):

*“Moreover, while both *ZNRF3* loss and *ZNRF3* RNA downregulation are significantly associated with higher grade disease, these features conferred risk of adverse outcomes independently of grade. This suggests that assessment of *ZNRF3* loss could help to identify a substantial fraction of patients who are at high risk of metastatic relapse, even amongst those with higher grade disease.*

*Thus in some sense the 10% overall rate of *ZNRF3* loss is misleading, because it encompasses all localized disease. Those 10% of cases are strongly enriched for men at the highest risk of metastatic relapse.”*

(ii) Though requested, no functional data regarding the role of ZNRF3 in any aspect of aggressive/metastatic behavior are provided.

Unfortunately, the COVID-19 pandemic and the associated shut-down of wet-lab activities has severely limited our ability to establish appropriate experimental models of *ZNRF3* loss. Nevertheless, we have added new data to the revised manuscript suggesting a functional mechanism through which *ZNRF3* loss may function. As shown in **Figure 5A**, *ZNRF3* loss is associated with substantial up-regulation of genes implicated in polycomb repressor complex (PRC) 1 and 2 function, which is strongly linked to adverse outcomes in both localized prostate cancer and mCRPC. mRNA levels of EZH2, EED, SUZ12, and CBX2 were significantly up-regulated in cases harbouring *ZNRF3* loss. Moreover, E2F/DREAM signaling was the most strongly up-regulated gene set in cases harbouring *ZNRF3* loss (**Figure 5B**)

(iii) As clarified in the revised manuscript and responses to the prior critique, there are 417 co-deleted with ZNRF3 on chromosome 22 and fully 29 of these were associated with biochemical relapse. Consequently – confidence that ZNRF3 independently drives adverse outcomes is not compelling and without confirmatory mechanistic data the conclusion is not well-supported.

The reviewer is partially correct: 29 genes were, indeed, associated with relapse. However, only 9/29 had RNA levels that were downregulated in cases where the gene showed monoallelic loss. As such, these 9 genes represent the strongest candidates to drive the aggressive phenotype. Moreover, the only one of these 9 gene with RNA abundance that was associated with relapse across four independent cohorts was *ZNRF3*. Moreover, three unique *ZNRF3*-associated features (monoallelic loss, RNA downregulation, and promoter hypermethylation) were each associated with adverse outcomes. As we noted in the revised manuscript, we cannot rule out the possibility that other genes in this region may contribute to aggressive disease. However, we believe that our data strongly support a role for *ZNRF3* in this regard.

We have revised the Discussion section to better reflect these conclusions (lines 657-665):

“Multiple lines of evidence support the hypothesis that ZNRF3 contributes to the clinical aggression observed in patients harbouring monoallelic chr22q12.1 loss. Of the 9 genes in the region that were significantly associated with metastasis-free survival when downregulated at the RNA level, low ZNRF3 RNA abundance was the only one that was prognostic across four independent validation cohorts. Moreover, the finding that tumours harbouring >1 ZNRF3 alterations (i.e. monoallelic loss, low RNA abundance, and/or 5’ hypermethylation) are significantly more aggressive than those harbouring 0-1 alterations strongly supports a role for ZNRF3 in promoting disease aggression. Nevertheless, we cannot exclude the possibility that other genes in the chr22q12.1 region contribute to this aggressive phenotype.”

(iv) The near-exclusive monoallelic copy loss and lack of any deleterious SNVs in ZNRF3 also suggests that ZNRF3 copy loss may not be the exclusive explanation for adverse outcomes in the setting of numerous other genomic alterations concurrent with ZNRF3 loss.

We concur with the reviewer. Our finding that ZNRF3 promoter hypermethylation is also (a) associated with lower ZNRF3 RNA abundance and (b) associated with disease progression and metastasis further supports a role for ZNRF3. Importantly, ZNRF3 promoter methylation was actually higher in cases with monoallelic ZNRF3 loss (despite the absence of one allele) than those lacking this CNA. This suggests a mechanism for silencing of the remaining allele.

We also note that single copy loss can confer an aggressive phenotype in localized prostate cancer; for example, loss of a single copy of NKX3-1 is associated with relapse following surgery or radiotherapy (PMID 22048240). Similar results have been shown for monoallelic loss of PTEN (PMID 17700571).

(v) ZNRF3 is not associated with adverse outcomes in all of the individual datasets comprising this analysis. The response is that ZNRF3 cooperates with other prognostic factors but this statement is not supported by strong mechanistic rationale.

The reviewer is correct; low ZNRF3 RNA abundance was not prognostic of progression-free survival in the Taylor/MSKCC cohort. However, ZNRF3 loss *was* associated with progression in this cohort. Unfortunately, no DNA methylation data are available for this cohort, and thus we cannot assess the impact of this feature – which was also prognostic of progression and metastasis in the TCGA and CPCG cohorts, respectively.

It is important to clearly note that in both cohorts for which metastasis-free survival was available, ZNRF3 loss and/or low ZNRF3 RNA abundance was prognostic of this endpoint, which, as noted, is a clinically validated surrogate of prostate cancer-specific mortality. Likewise, despite the low event rate, ZNRF3 loss was directly prognostic of prostate cancer-specific mortality in TCGA.

Thus, while we cannot rule out that other genes/mutations may interact with ZNRF3, we believe we have demonstrated that ZNRF3 loss/RNA downregulation/hypermethylation clearly affects the most clinically meaningful survival endpoints.

(vi) As noted in the prior review, there are very few cases with biallelic loss of ZNRF3 and the authors appeared to suggest that ZNRF3 is functioning as a haplo-insufficient tumor/metastasis suppressor. The authors now agree with this possibility. However, no data are shown that clearly confirms this hypothesis.

To further investigate other somatic features that may affect clinical aggression in the context of ZNRF3, we performed a differential DNA methylation analysis using 450K microarray data for both TCGA and CPCG. In both cohorts, the 5' promoter region of ZNRF3 was significantly hypermethylated in cases with low ZNRF3 RNA abundance. Strikingly, methylation of this locus was also *higher* in cases harbouring monoallelic ZNRF3 loss and ZNRF3 hypermethylation was associated with increased risk of metastatic relapse in CPCG and with disease progression in TCGA (**Figure 5C-D, Supplementary Figure 14**). Moreover, patients who had >1 aberrant ZNRF3-associated feature (loss, RNA downregulation, hypermethylation) were at significantly higher risk of relapse than those with only one alteration. These data have been added to the revised manuscript (**lines 530-549**):

“The vast majority of ZNRF3 losses are monoallelic. In TCGA, for example, 7/65 (10.8%) ZNRF3 losses are biallelic, while across both mCRPC cohorts, 7/157 (4.5%) ZNRF3 losses are biallelic. This suggests that ZNRF3 may function as a haploinsufficient tumour suppressor in prostate cancer. Alternatively, other mechanisms – including epigenetic silencing – may contribute to the downregulation of ZNRF3 RNA observed in aggressive localized disease. To that end, we next analysed global DNA methylation patterns in localized prostate cancers associated with ZNRF3 RNA downregulation. In both the CPG and TCGA cohorts, the most significantly differentially methylated CpG was located in the ZNRF3 5’ promoter region (probe ID: cg11986861; Figure 4A & S11A-B). Methylation of this CpG was increased by 1.4-fold and 1.7-fold in CPG and TCGA cases with low ZNRF3 RNA abundance, respectively, and was inversely correlated with ZNRF3 RNA abundance (CPG: $\rho = -0.406$, $p = 3.19 \times 10^{-9}$; Figure 4B; TCGA: $\rho = -0.496$, $p = 1.12 \times 10^{-31}$; Figure S11C). Despite the loss of one ZNRF3 allele, this CpG was also significantly hypermethylated in tumours with ZNRF3 loss (CPG: $p = 9.16 \times 10^{-3}$; TCGA: $p = 2.48 \times 10^{-3}$, Mann-Whitney U test; Figure 4C-D). ZNRF3 hypermethylation was associated with increased risk of metastatic relapse (HR: 2.18, 95% CI: 1.06 – 4.05; $p = 3.47 \times 10^{-2}$; Wald test) and shorter progression-free survival (HR: 2.19, 95% CI: 1.45 – 3.31; $p = 1.83 \times 10^{-4}$; Wald test) in CPG and TCGA, respectively. Moreover, patients in both the CPG and TCGA cohorts who harboured more than one ZNRF3-associated feature (i.e. monoallelic loss, low RNA abundance, and/or hypermethylation) were at significantly higher risk of adverse outcomes than those who harboured one or fewer aberrant features (Figure 4E-F).”

5. The analyses failed to identify several gene/genomic alterations that have been clearly associated with adverse outcomes in many prior studies such as PTEN loss and state that the issue may be with the “reduced power of genome-wide studies such as this one relative to candidate gene analyses..”. However, it is notable that the present study is not ‘genome-wide’ it is an analysis of 113 preselected genomic alterations.

The reviewer is correct. PTEN loss was not associated with metastatic relapse in CPG, although it was associated with biochemical relapse. We have added this to the revised manuscript (**lines 401-402**):

“PTEN and RB1 loss were not prognostic of metastatic relapse (although PTEN loss was associated with biochemical relapse in the CPG cohort; $p = 0.035$, log-rank test).”

We apologize for the lack of clarity regarding the term ‘genome-wide’. The list of 113 preselected mutations analysed herein was derived from two studies of prostate cancer whole genome sequencing (Fraser et al, Nature, 2017 and Quigley et al, Cell, 2018). Our intention was to differentiate between studies of individually selected genes and those informed by an unbiased genomics approach. We have clarified this in the revised discussion, as stated on **lines 618-622**:

“This may be due to the reduced power of multi-gene studies (based on genome-wide surveys) such as this one relative to candidate gene analyses, which have previously

suggested a role for these tumour suppressors in event-free survival and metastatic relapse⁶⁴. Alternatively, our findings may, in part, reflect a unique biology associated with the more advanced disease represented in TCGA vs. CPGC.”

6. Minor point – the authors state that ZNRF3 could serve as a predictive biomarker for porcupine inhibitors. However, it is not clear that any porcupine inhibitors are approved for any indication.

The reviewer is correct; to our knowledge, porcupine inhibitors are not approved for any indication. They are, however, under investigation for WNT- and NOTCH-driven cancers (e.g. NCT01351103).

7. Overall, the impact of this study would be substantially increased if the ongoing preclinical studies alluded to in the authors responses involving prostate cancers with/without ZNRF3 loss demonstrated features associated with aggressive/metastatic behaviors and/or differential responses to WNT pathway antagonists.

While we agree with the reviewer that mechanistic experiments would help to expand the impact of the study, we highlight the substantial new data added in this revision:

- Differential DNA methylation analysis demonstrating *ZNRF3* 5' promoter hypermethylation in TCGA and CPGC patients harbouring low *ZNRF3* RNA
- *ZNRF3* hypermethylation in tumours harbouring monoallelic *ZNRF3* loss, suggesting epigenetic silencing of the remaining allele;
- Upregulation of multiple genes implicated in Polycomb Repressive Complex-1 and -2 signaling (e.g. *EZH2*, *EED*, *SUZ12*, *CBX2*) in localized cancers harbouring *ZNRF3* loss, demonstrating a mechanistic link between *ZNRF3* loss and an determinant of aggressive prostate cancer (i.e. *PRC1/2*).

We sincerely thank the reviewer for their careful examination of our manuscript, and we hope that with these additional new data – as well as the refinement to text – that the manuscript will now be acceptable for publication.

REVIEWERS' COMMENTS

Reviewer #3 (Remarks to the Author):

This is a second revision of the manuscript: "Somatic Driver Mutation Prevalence in 1,844 Prostate Cancers Identifies ZNRF3 Loss as a Predictor of Metastatic Relapse"

The major issue raised in the prior critique centered on a lack of functional data supporting the role of ZNRF3 - the gene/locus focused on by the authors - and prostate cancer, including a potential role as a haploinsufficient tumor suppressor.

In the response, the authors have emphasized a viewpoint that ZNRF3 loss - though rare across prostate cancer - none-the-less represents an important prognostic feature. No further functional data were provided.

It is then a matter of opinion regarding the impact of the present study on the field. Though the manuscript is well-written and the replies to the prior critiques are thoughtful, without confirmatory functional data, it is the opinion of this reviewer that the present data do not represent a major advance in the field.

No new data are provided. The method, though statistically rigorous, is not novel. There are numerous other genes in the locus that associate with outcome. ZNRF3 loss is not common (note that ZNRF3 loss occurs in 14/206 patients in the CPGC cohort - Fig 5D? - Metastasis-free survival plot of only ZNRF3 appears to overlap with neutral - Fig 5C?) and the designation of a gene as a haploinsufficient tumor suppressor has important implications that are not rigorously supported by functional data.

REVIEWERS' COMMENTS

Reviewer #3 (Remarks to the Author):

This is a second revision of the manuscript: "Somatic Driver Mutation Prevalence in 1,844 Prostate Cancers Identifies ZNRF3 Loss as a Predictor of Metastatic Relapse"

The major issue raised in the prior critique centered on a lack of functional data supporting the role of ZNRF3 - the gene/locus focused on by the authors - and prostate cancer, including a potential role as a haploinsufficient tumor suppressor.

In the response, the authors have emphasized a viewpoint that ZNRF3 loss - though rare across prostate cancer - none-the-less represents an important prognostic feature. No further functional data were provided.

We sincerely thank the reviewer for their thoughtful comments throughout the review period. While no functional (i.e. experimental) data have been provided, we emphasize that new data have been added following the first round of reviews, particularly with regards to the hypermethylation of the ZNRF3 promoter and the association of this hypermethylation – both alone and in the context of ZNRF3 loss – on clinical outcomes. We feel these new data further support our hypothesis that alterations in ZNRF3 copy number, RNA abundance, and/or methylation could be used to identify men at risk for metastatic relapse following curative intent therapy for localized prostate cancer.

It is then a matter of opinion regarding the impact of the present study on the field. Though the manuscript is well-written and the replies to the prior critiques are thoughtful, without confirmatory functional data, it is the opinion of this reviewer that the present data do not represent a major advance in the field.

While we feel that our study provides substantial evidence for a novel prognostic biomarker in localized prostate cancer (with particular importance to a clinically-validated surrogate endpoint of prostate cancer-specific mortality), we appreciate that the reviewer does not share this opinion.

No new data are provided. The method, though statistically rigorous, is not novel. There are numerous other genes in the locus that associate with outcome. ZNRF3 loss is not common (note that ZNRF3 loss occurs in 14/206 patients in the CPCG cohort - Fig 5D? - Metastasis-free survival plot of only ZNRF3 appears to overlap with neutral - Fig 5C?) and the designation of a gene as a haploinsufficient tumor suppressor has important implications that are not rigorously supported by functional data.

We appreciate this comment, and we agree with the reviewer that some clarification is required regarding the potential role of ZNRF3 as a haploinsufficient tumour suppressor.

First, we would like to clarify that the 14/206 (6.7%) proportion quoted above refers to those CPCG patients for whom RNA abundance data are available. The overall proportion in CPC-GENE is 31/375 (8.3%; see Figure 3C, for example). Moreover, we have provided new data (Figure S4C) showing that ZNRF3 loss is significantly correlated with higher tumour grade. Because the CPC-GENE cohort consists of only patients with intermediate risk disease – about 1/3 of which are Gleason 6 with high PSA – we would expect to see a lower proportion of these cases harbouring ZNRF3 loss. Indeed, this is precisely what we observe; in the TCGA cohort, which consists of a higher average tumour grade, 58/483 (12%) of patients harboured ZNRF3 loss. We have added the following text to the Discussion (lines 613-615):

“Importantly, ZNRF3 loss is associated with higher tumour grade but provides prognostic value independently of grade and other clinical prognostic factors.”

With regards to Figure 5C (overlap of ZNRF3 loss with neutral), the reviewer is correct. We would note, however, that CNAs in CCND1 alone are also not particularly informative with respect to metastatic relapse. However, the prognostic impact of having both CCND1 and ZNRF3 CNAs is very large, with all but 1/8 patients experiencing a metastatic relapse within ~6 years.

In addition, we note that ZNRF3 loss and 5' promoter hypermethylation can both contribute to decreased ZNRF3 RNA abundance (Figures 4A-D) and that patients with both loss and hypermethylation of ZNRF3 (26/201 patients in CPCG; 129/488 patients in TCGA) have significantly worse outcomes than those with only loss or hypermethylation (Figure 4E-F). We have added the following text to the Discussion (lines 638-641):

“While ZNRF3 loss is comparatively rare in localized disease, the apparent interaction of at least two mechanisms of ZNRF3 silencing (loss, hypermethylation) suggests that the detection of loss alone, may underestimate the impact of this gene as a predictor of adverse outcomes.”

Our new data with regards to hypermethylation of ZNRF3 (Figure 4), even in tumours with ZNRF3 loss, suggests that, in fact, ZNRF3 may not be a haploinsufficient tumour suppressor but instead that epigenetic silencing of the remaining allele may contribute to the differential clinical outcomes in these patients. To that end, we have removed the line suggesting that ZNRF3 may function as a haploinsufficient tumour suppressor (line 526): *“This suggests that ZNRF3 may function as a haploinsufficient tumour suppressor in prostate cancer.”*

Furthermore, we have added new text to the Discussion with regards to epigenetic silencing (lines 670-672):

“These data are consistent with the hypothesis that monoallelic loss of ZNRF3 and epigenetic silencing of the remaining allele may each contribute to a reduction in ZNRF3 RNA abundance and increased tumour aggression.”

As a final comment, we would like to sincerely thank the reviewer (and, indeed, all the reviewers) for their efforts on this manuscript. While we may have some differences with respect to our individual interpretations of the study results, the reviewer's comments have dramatically improved the quality of the manuscript. At every turn, this has forced us to question our interpretations, in many cases, we have reinterpreted those data based on the thoughtful and insightful comments we have received.

Many thanks!